# COMPETITIVE MULTI-AGENT DELEGATION FOR LLM REASONING

## ABSTRACT

Large Language Models (LLMs) have shown impressive capabilities in natural language generation, yet they remain limited in complex and multi-step reasoning. We propose COMMAND: COMpetitive Multi-AgeNt Delegation, a framework where a principal LLM assigns tasks to multiple agent LLMs. Agents compete in an environment where utilities depend on both their internal confidence and the principal's evaluation, incentivizing answers that are higher-quality and better aligned with the principal. We establish theoretical guarantees demonstrating that, under fair comparison, multi-agent systems such as COMMAND provably outperform their single-agent counterparts. Moreover, each agent, via online learning, achieves sublinear regret and its average policy will converge to a Nash equilibrium. Empirical evaluations on multiple benchmarks demonstrate that COMMAND yields significant improvements in factual accuracy.

## 1 INTRODUCTION

Large Language Models (LLMs) have demonstrated impressive capabilities across a wide range of natural language tasks (Zheng et al., 2023; Achiam et al., 2023; Koike et al., 2024). However, they still face challenges in complex reasoning scenarios that require multi-step reasoning, where problems cannot be solved in a single leap but must be decomposed into subproblems and integrated through intermediate results (Mirzadeh et al., 2024; Stechly et al., 2024). For example, solving mathematical problems often involves a chain of deductions in which each step depends on the previous one (Imani et al., 2023). A prominent line of work focuses on inference-time prompting strategies. These methods guide the model to generate diverse reasoning paths, such as chain-of-thought prompting (Wei et al., 2022) or sampling multiple rationales (Guan et al., 2025), and then selects the most reliable outcome using a reward model or verifier (Lightman et al., 2024). However, these methods treat each reasoning path in isolation, without principled mechanisms for interaction or refinement, and their effectiveness is limited by the inherent capacity of each model (Sprague et al., 2024; Xu et al., 2024; Stechly et al., 2024; Wang et al., 2022).

Inspired by human reasoning, recent methods encourage models to refine their outputs through self-reflection, critique, and iterative improvement (Madaan et al., 2023; Cheng et al., 2024). In parallel, ensemble methods involve multiple LLMs engaging in debate, feedback exchange, or negotiation to enhance answer quality (Huang et al., 2024b; Chen et al., 2025; Wang et al., 2024a). Despite the progress of these approaches, they lack formal guarantees that iterative refinement or ensemble methods will improve reasoning quality. Moreover, ensemble methods depend on each model having substantial capacity, which limits their effectiveness when individual models are weak.

We propose COMpetitive Multi-AgeNt Delegation (COMMAND), a training-free, game-theoretic framework designed to improve LLM reasoning abilities *without* access to additional fine-tuning, parameter updates, or task-specific retraining. In this framework, multiple agent LLMs independently generate candidate answers to a given task and select one for submission. The principal LLM then evaluates each submission. The utility of each agent is determined by its internal utility over the answer and its relative ranking given by the principal, so each agent must balance its internal preferences with the likelihood of receiving a favorable ranking from the principal. This mutual evaluation structure ensures that agents are incentivized to improve their answers in ways that align with the principal's evaluation criteria. To implement this framework, we apply the online mirror descent algorithm to iteratively update each agent's policy, enabling the system to converge to equilibrium.

The design of COMMAND is also inspired by concepts from biology, where competition among diverse entities drives adaptation and the emergence of more effective answers (Endler, 1986; Albadr et al., 2020). From an economic perspective, market competition illustrates how multiple actors striving to maximize their own utility can collectively enhance overall efficiency (Podolny, 1993; Gupta et al., 2016). This collective dynamic offers a scalable and label-free approach to building more trustworthy LLM systems without golden answers or manually annotated data.

To summarize, the main contribution of our work is as follows:

- We introduce COMMAND, a competitive multi-agent framework for enhancing LLM reasoning. In this framework, a principal LLM engages multiple agent LLMs in a game-theoretic setting, where agents compete to propose high-quality answers that align with the principal's evaluation.

- We provide theoretical guarantees showing that multi-agent framework of COMMAND improve over single-agent counterpart. Moreover, when agents update their policies using online mirror descent, the system achieves sublinear regret and the average policy converges to a Nash equilibrium.

- We conduct comprehensive empirical evaluations on multiple reasoning benchmarks, demonstrating significant gains in factual accuracy.

## 2 METHODOLOGY

### 2.1 MOTIVATION

LLMs are observed to fall short on complex tasks such as mathematical reasoning, multi-step planning, and commonsense reasoning (Mirzadeh et al., 2024; Stechly et al., 2024; Kwon et al., 2024). One promising direction to address these limitations is to encourage a *single* LLM to think like humans by generating multiple reasoning paths and selecting the most plausible answer, analogous to how humans tackle complex problems through diverse strategies (Yao et al., 2023). These are often coupled with a verifier model or selection criterion to identify and retain the most consistent or accurate response (Guan et al., 2025; Cobbe et al., 2021b). Specifically, for a given task $t \in \mathcal{T}$, an LLM may generate multiple candidate answers $\{\omega_1, \ldots, \omega_K\}$ either through repeated sampling or by encoding diverse reasoning paths in prompts (Guan et al., 2025; Cobbe et al., 2021b). The final answer is selected by maximizing utility $U : \mathcal{A} \to \mathbb{R}$, where $\mathcal{A}$ denotes the set of admissible answers, i.e., the LLM output space. The utility $U$ can be instantiated as an external reward model or derived from internal evaluations such as consistency or factuality checks (Zhang et al., 2024; Guan et al., 2025). That is, the selected answer is given by $\omega^* = \arg\max_{\omega \in \{\omega_1, \ldots, \omega_K\}} U(\omega)$.

However, this *single*-LLM approach faces two main challenges. First, answer selection can be biased and unreliable, since reward models or self-evaluations may not capture true task quality (Zheng et al., 2024; Wang et al., 2024b). Second, candidate answers are usually treated in isolation, ignoring potential complementarities or cross-validation among them. To address these issues, we propose to leverage *multi-agent* LLMs, where multiple models act as competing agents and provide diverse perspectives that mitigate bias and improve robustness in the selection process.

### 2.2 COMPETITIVE MULTI-AGENT DELEGATION GAME

We reinterpret the reasoning-and-selection process, where an LLM generates candidate responses that are later filtered into a final output, as a *delegation game*. In this game-theoretic view, a principal delegates tasks to self-interested agents with potentially misaligned preferences and then selects from their responses to achieve desirable outputs (Fershtman et al., 1991; Frankel, 2014; Guo, 2016). Building on this perspective, we model a delegation game where agents, each following a distinct reasoning strategy, submit candidate responses, and a principal acts as centralized evaluator to rank them. We term this setup the COMpetitive Multi-AgeNt Delegation (COMMAND) game, highlighting how principal feedback induces competition among heterogeneous multi-agents.

Specifically, for each task $t \in \mathcal{T}$ and each agent $i \in [N]$, the agent generates a set of candidates $\mathcal{A}_i = \{\omega_{i1}, \ldots, \omega_{iK}\} \subseteq \mathcal{A}$, then selects a submission $a_i \in \mathcal{A}_i$ according to its internal utility $U_{y_i} : \mathcal{A} \to \mathbb{R}$. This utility is operationalized via *self-consistency* (Wang et al., 2022): $U_{y_i}(a) := \mathbb{P}_i(a|t) \approx$

$\frac{1}{K}\sum_{k=1}^{K}\mathbb{I}[\omega_{ik} = a]$, where the probability is estimated by the empirical frequency of $a$ among the sampled responses. Intuitively, higher utility is assigned to answers the agent would generate more consistently. Then, the principal aggregates the submissions $\{a_1, \dots, a_N\}$ and evaluates them via a global utility $U : \mathcal{A} \to \mathbb{R}$ aligned with user preferences. It is shaped by implicit or explicit user instructions via the prompts, with the principal acting as a proxy for the user, selecting the response that best aligns with the user's intended objective. After evaluation, each agent $i$ receives a feedback $r_i \in \mathbb{R}$ determined by the relative ranking of $a_i$ (e.g., $r_i = +1$ for top rank, $r_i = -1$ for bottom rank, and intermediate values otherwise). This design encourages agents to explore diverse reasoning paths, but it also increases the risk of misalignment between their local objectives and the principal's utility. To reconcile the two, each agent $i$'s reward is defined as,

$$U_i(a_i) = r_i \cdot U_{y_i}(a_i),$$

which combines its internal preference with the ranking feedback from the principal. The ranking-based mechanism creates structural tension: agents must balance their internal reasoning with the principal's evaluation, leading them to refine strategies that improve both their individual quality and their relative standing. The overall process of COMMAND is illustrated in Figure 1.

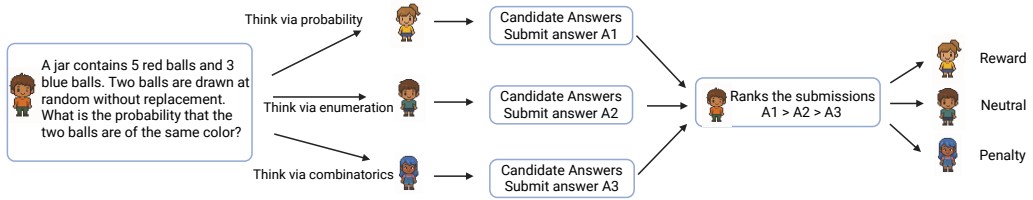

Figure 1: Overview of COMMAND: multiple agents propose answers along distinct reasoning paths and receive utility from the principal's ranking feedback. This setup incentivizes high-confidence outputs while promoting alignment with the principal.

### 2.3 IMPLEMENTATION OF COMMAND

In the implementation, COMMAND consists of the following steps. (i) Given a task $t \in \mathcal{T}$, each agent generates a set of candidate answers. (ii) Each agent selects one answer to submit, according to a policy that defines a probability distribution over its candidates, estimated via self-consistency from repeated sampling. (iii) The principal evaluates and ranks the submitted answers according to its own utility. (iv) This ranking serves as feedback, where each agent receives a scalar utility based on the relative position of its answer. (v) Each agent updates its policy based on this feedback, with the goal of improving future performance while competing against others.

For agent $i$, we define a policy $\pi_i$ as a probability distribution over its candidate set $\mathcal{A}_i$, where $\pi_i(a)$ denotes the probability of selecting an answer $a \in \mathcal{A}_i$. The agent's objective is to adaptively update this policy to increase its expected cumulative utility, given the outcomes of prior interactions. We adopt a mirror descent update rule, which adjusts the policy by shifting probability mass toward answers with higher observed utility, while maintaining exploration through regularization (Duvocelle et al., 2023; Jacob et al., 2022). Specifically, mirror descent provides a principled framework for updating distributions over actions, and in our setting corresponds to an exponential weighting scheme based on utility feedback (Shalev-Shwartz, 2012). When using negative entropy as the mirror map, the resulting update recovers the classical Hedge algorithm from online learning (Littlestone & Warmuth, 1994). The full procedure is described in Algorithm 1.

## 3 THEORETICAL GUARANTEES

### 3.1 PROVABLE IMPROVEMENTS VIA MULTI-AGENT DELEGATION

We establish that multi-agent delegation can yield better performance than the single-agent setting. To enable a fair comparison, we ensure both settings have equal total access to candidate answers. In the single-agent case, the agent draws $k$ answers from a distribution $D$ to form its candidate set. In the multi-agent case, we consider $N \geq 2$ agents, where each agent $i$ independently draws $k_i$

---

**Algorithm 1** Implementation of COMMAND with Mirror Descent

---

1: **Initialize:** For each agent $i \in [N]$, set candidate set $\mathcal{A}_i$, initialize utility estimates $U_i^0(a) \leftarrow 0$ and policy $\pi_i^0(a)$ for all $a \in \mathcal{A}_i$, choose learning rate $\eta > 0$.
2: **for** each round $t = 1, 2, \dots$ **do**
3:     **for** each agent $i \in [N]$ (in parallel) **do**
4:         Sample answer $a_i^t \sim \pi_i^t$ and submit $a_i^t$ to principal
5:     **end for**
6:     Principal ranks the submitted answers and provides feedback $r_i^t$
7:     **for** each agent $i \in [N]$ **do**
8:         **for** each candidate answer $a \in \mathcal{A}_i$ **do**
9:             Update utility:
$$U_i^t(a) \leftarrow U_i^{t-1}(a) + r_i^t U_{y_i}(a, a_{-i}^t)$$
10:         **end for**
11:         Compute policy: $\pi_i^t(a) \propto \exp\left\{\eta U_i^{t-1}(a)\right\}, \quad \forall a \in \mathcal{A}_i$
12:     **end for**
13: **end for**

---

samples from $D$, with the same total number of samples: $\sum_{i=1}^N k_i = k$. We assume all agents share the same internal utility function, $U_{y_i} = U_y$, and that the principal applies the same evaluation function $U$ across both settings. Under these conditions, any performance improvement arises solely from delegation rather than unequal information.

**Assumption 1.** *(i) Pareto-optimal play. If an agent has two candidate answers $\omega$ and $\omega'$ such that the principal's utility $U(\omega) \geq U(\omega')$ and the agent's utility $U_{y_i}(\omega) \geq U_{y_i}(\omega')$, with at least one inequality being strict, then the agent does not submit $\omega'$ to the principal.*

*(ii) Symmetric agents. All agents generate the same number of candidate answers, and the utility of each answer follows the same distribution $D$.*

*(iii) Non-negative alignment. The principal's utility $U(\cdot)$ are not negatively correlated with any agent's utility $U_{y_i}(\cdot)$, i.e., $\mathrm{Corr}\big(U(\omega), U_{y_i}(\omega)\big) \geq 0$, for $\omega \sim D$.*

Part (i) of Assumption 1 enforces Pareto-optimal play: if one answer is at least as good for both the agent and the principal, and strictly better for at least one, then the agent will not choose the inferior one. Part (ii) ensures that all agents operate under comparable conditions. Each agent generates the same number of candidate answers and submits one to the principal, so no agent gains an advantage from producing more options. By using LLMs of similar capacity with identical sampling procedures, the distribution of generated candidates is symmetric across agents. Part (iii) rules out adversarial behavior by ensuring that when the principal values an answer, agents do not systematically devalue it. A detailed discussion of the necessity of Assumption 1 is provided in Appendix A.4. Additionally, we empirically verify parts (i) and (iii) in Section 4.3.

To evaluate the efficiency of mechanisms in strategic settings, we compare their outcomes with the expected value of the optimal result, denoted as $\mathbb{E}[U_{\max}]$, where $U_{\max}$ represents the principal's utility from the best candidate answer among those generated by the agent. A mechanism $M$, under agent strategies $\sigma$, is said to be $(\rho, \gamma)$-approximate if its expected outcome satisfies $\rho\mathbb{E}[U_{M,\sigma}] + \gamma \geq \mathbb{E}[U_{\max}]$, where $\mathbb{E}[U_{M,\sigma}]$ denotes the principal's expected utility from the answer selected by $M$ under strategies $\sigma$, and $\rho$ and $\gamma$ are the multiplicative and additive approximation factors, respectively. In our setting, we focus on prior-independent mechanism, where the mechanism has no prior knowledge of the distributions from which the agents' answers are drawn. Additionally, we adopt an incomplete information framework among agents: while agents can be aware of the principal's utility function, they do not observe each other's submitted answers but only observe the ranking feedback returned by the principal.

**Theorem 1.** *Consider a single-agent problem $P$ and its multi-agent correspondence $P'$ with $N$ agents. Then under Assumption 1, we have,*

    *(a) For any mechanism $M$ under $P$, there exists a multi-agent single-proposal mechanism $M'$ under $P'$ such that, at the Nash equilibrium of each mechanism, $U(M') \geq U(M)$, where $U(\cdot)$ denotes the principal's utility function.*

*(b) When each agent $i$ generates candidate answers independently and the principal utility of these candidates follow $\mathcal{U}[-1,1]$, and the agent is willing to tolerate a utility loss of at most $2\varepsilon$ relative to its optimal candidate. That is, instead of always selecting the answer that maximizes its own utility $U_i^{\max}$, the agent may strategically submit any answer whose utility is at least $U_{i,\max} - 2\varepsilon$. As a result, a $2\varepsilon$-approximate Bayes-Nash equilibrium can be achieved and the expected utility attained by the principal satisfies $\mathbb{E}[U_{M',\sigma'}] = \mathbb{E}[U_{\max}]$, where $\varepsilon \leq 1 - e^{-\frac{N^2}{2(N-1)^2}}$.*

Part (a) of this theorem shows that the principal can obtain a higher-utility answer by recruiting more comparable agents. Intuitively, when multiple agents each submit one answer, the principal has more chances to receive a better answer, as independent judgments may increase the likelihood of a better outcome. Part (b) further shows that, if each agent is not purely self-interested and is willing to consider answers with utility at least $U_{i,\max} - 2\varepsilon$, then the principal can obtain the best answer with utility $\mathbb{E}[U_{\max}]$. Unlike prior work that assumes independent utilities (Shin et al., 2023), our setting involves a shared problem between the principal and agents, making independence unrealistic. We therefore extend the analysis to positively correlated utilities (Assumption 1), which more accurately capture the structural alignment between LLM agent and principal objectives.

## 3.2 REGRET ANALYSIS

We analyze the learning dynamics of our algorithm in the multi-agent setting and establish regret guarantees for each agent. Regret is evaluated with respect to each agent's own utility and measures the gap between the utility the agent actually obtains and the utility it could have obtained by following the best fixed policy in hindsight. Specifically, we show that each agent achieves sublinear regret over time when running Algorithm 1, so its cumulative utility asymptotically matches that of its best fixed policy.

For agent $i$ and any candidate answers $a \in \mathcal{A}_i$, we define the regret after $T$ rounds as the average difference between the cumulative utility the agent would have received by consistently selecting an answer $a$, and the utility actually obtained by following the sequence of mixed strategies $\{\pi_i^t\}_{t=1}^T$ (Cai & Zheng, 2023). Formally, the regret is defined as

$$R_i^T(a) = \max_\pi \left\{ \sum_{t=1}^T [U_i(\pi, a_{-i}^t) - U_i(\pi_i^t, a_{-i}^t)] \right\},$$

where $a_{-i}^t$ denotes the answers chosen by all agents other than $i$ at round $t$, and $u_i(\pi_i^t, a_{-i}^t) = \sum_{a' \in \mathcal{A}_i} \pi_i^t(a') U_i(a', a_{-i}^t)$ is the expected utility of agent $i$ under their mixed strategy $\pi_i^t$. The regret quantifies, for each fixed answer $a$, how much worse the agent performed on average compared to always selecting that answer. The agent is said to have *no regret* if $\max_{a \in \mathcal{A}_i} R_i^T(a)/T \to 0$ as $T \to \infty$ (Bubeck et al., 2012). More generally, one can express the cumulative regret with respect to any fixed mixed strategy $\pi$, in which case the following upper bound holds:

**Theorem 2.** *Let agent $i$ follow Algorithm 1 with learning rate $\eta = 1/\sqrt{T}$. Then for any policy $\pi$, the cumulative regret over $T$ rounds satisfies*

$$\sum_{t=1}^T \left[ U_i(\pi, a_{-i}^t) - U_i(\pi_i^t, a_{-i}^t) \right] \leq \left( \frac{1}{4} + D_{\mathrm{KL}}(\pi \,\|\, \pi_i^0) \right) \sqrt{T}.$$

Theorem 2 shows that when each agent follows Algorithm 1 with learning rate $\eta = 1/\sqrt{T}$, the cumulative regret after $T$ rounds is upper bounded by $\mathcal{O}(\sqrt{T})$. The factor in the bound depends on two components: a fixed coefficient $1/4$ from the optimization dynamics, and the KL divergence between the comparator policy and the initial policy. As a result, the average regret $\sum_{t=1}^T \left[ U_i(\pi, a_{-i}^t) - U_i(\pi_i^t, a_{-i}^t) \right]/T$ vanishes as $T \to \infty$, which ensures that each agent learns to perform competitively over time. This guarantee is consistent with standard results in online learning and mirror descent algorithms (Cai & Zheng, 2023; Jacob et al., 2022).

## 3.3 CONVERGENCE GUARANTEES

We now analyze the convergence behavior of agents to a Nash equilibrium. In our setting, a Nash equilibrium corresponds to a stable outcome in which each agent adopts the best policy over candi-

date answers that maximizes its expected utility, given the fixed policies of all other agents (Kreps, 1989; Holt & Roth, 2004). Formally, let $\pi = (\pi_1, \ldots, \pi_N)$ denote the joint policy profile, where each $\pi_i$ is a probability distribution over the candidate set $\mathcal{A}_i$ of agent $i$. Then $\pi^* = (\pi_1^*, \ldots, \pi_N^*)$ is a Nash equilibrium if for all $i \in N$ and any alternative policy $\hat{\pi}_i$,

$$U_i(\pi_i^*, \pi_{-i}^*) \geq U_i(\hat{\pi}_i, \pi_{-i}^*).$$

At equilibrium, no agent can improve its expected utility by changing its policy alone.

**Theorem 3.** *For any $T \in \mathbb{N}$, $\eta > 0$, and $\delta \in (0, 1)$, define the quantity*

$$\xi^T(\delta) := \frac{\sum_{i=1}^N R_i^T}{T} + N\sqrt{\frac{8}{T} \log\left(\frac{N \max_i |\mathcal{A}_i|}{\delta}\right)}.$$

*For $N$-agents delegation games, upon running Algorithm 1 for any $T$ iterations with learning rate $\eta > 0$, the average policies $\bar{\pi}_i^T = \frac{1}{T} \sum_{t=1}^T \pi_i^t$ of each agent form a $\xi^T(\delta)$-approximate Nash equilibrium with probability at least $1 - \delta$, for any $\delta \in (0, 1)$.*

Theorem 3 establishes that the empirical average of the policies produced by Algorithm 1 converges toward equilibrium behavior. Specifically, the average strategy profile $(\bar{\pi}_1^T, \ldots, \bar{\pi}_N^T)$ is guaranteed to be a $\xi^T(\delta)$-approximate Nash equilibrium with high probability. The error term $\xi^T(\delta)$ decomposes into two parts: (i) the cumulative regret $\sum_i R_i^T / T$, which vanishes sublinearly under Algorithm 1, and (ii) a concentration term of order $O(\sqrt{\log(N \max_i |\mathcal{A}_i|/\delta)/T})$ arising from standard martingale inequalities. Together, these imply that the approximation error decays at the rate $O(1/\sqrt{T})$, ensuring that no player can improve her long-run utility by more than $O(1/\sqrt{T})$ through unilateral deviation.

The equilibrium reflects the outcome of repeated learning dynamics where agents iteratively adapt their policies to better align with the principal's feedback while preserving their own reasoning-based preferences. In this sense, each agent's policy converges to a distribution over answers that balances two forces: being favored by the agent's internal utility and being ranked highly by the principal. The result generalizes the convergence guarantees of Jacob et al. (2022) from two-player zero-sum games to general multi-agent delegation settings.

## 4 EXPERIMENTS

### 4.1 EXPERIMENT SETUP

**Models**. We adopt four open-sourced instruction-following LLMs to serve as agents in our delegation framework: Mistral-8B-Instruct (Mistral (2024)), Zephyr-7B-Beta (Tunstall et al. (2023) ), Phi-3-Mini-4K-Instruct (Abdin et al. (2024)) and Falcon-7B-Instruct (Almazrouei et al. (2023)). These models have comparable parameters and are matched in capacity, satisfying Assumption 1, while contributing architectural and training diversity to the agent pool. We use LLaMA-2-7B-Instruct (Touvron et al. (2023b)) as the principal because of its good alignment and consistent performance on all of our reasoning benchmarks. Unless otherwise specified, we set the learning rate $\eta = 0.1$ and 20 maximum iteration number for all experiments.

**Implementation Details.** In the candidate answer generation stage, we augment each agent with Monte Carlo Tree Search (MCTS) , performing 16 roll-outs with 5 maximum depth. In the delegation stage, a Llama2-7b model (Touvron et al., 2023a) served as the principal, providing feedback via best-path masking and consistency checks to the agents. The prompts used throughout the system adhered to the format described in the work by Guan et al. (2025).

**Baselines.** We compare our delegation-based reasoning framework against three strong and representative baselines:

- **Few-shot Chain-of-Thought** (Wei et al., 2022) is a method that prompts a LLM with a few in-context exemplars illustrating intermediate reasoning steps before the final answer. This baseline reflects the model's standalone reasoning ability without delegation.
- **rStar** (Guan et al., 2025) is a self-play approach that improves reasoning through a generation and discrimination process. It first generates multiple reasoning paths and then

uses a discriminator LLM to filter answers. In our implementation, the principal LLM fills in missing steps given earlier steps and scores candidate answers. We adopt a simplified variant that only verifies and scores the agents' submitted answers.

- To compare against a setting where the principal acts alone (without delegation), we also report the results selected by the principal in the final iteration; we refer to this setting as **Principal**.

**Datasets**. We conduct our evaluations using reasoning datasets: GSM8K, MATH, and GSM-Hard (Cobbe et al., 2021a; Lightman et al., 2024; Gao et al., 2022). Each dataset presents unique challenges, allowing us to evaluate the proposed delegation framework across different types of reasoning skills and linguistic variation. To manage compute, we evaluate 300 sampled problems from the official test splits of MATH and GSM8K, and for GSM-Hard, we evaluate on 320 test questions. Details are summarized below.

- MATH (Lightman et al. (2024)) is a dataset of challenging competition-level mathematics problems ranging from high school to early college level. It covers diverse topics such as algebra, geometry, probability, and calculus, requiring models to demonstrate advanced mathematical reasoning and symbolic manipulation skills.
- GSM8K (Cobbe et al. (2021b)) is a benchmark for grade-school math problems. It composed of 8.5K high-quality word problems that require multi-step arithmetic reasoning. The test set is about 1.3k. The dataset evaluates how well models can perform numerical computations and logical deductions in everyday language scenarios.
- GSM-Hard (Gao et al. (2022)) is a hard version of GSM8K math reasoning dataset. It replace the numbers in the questions of GSM8K with larger numbers that are less common.

## 4.2 EXPERIMENT RESULTS

**COMMAND Improves Accuracy.** Results in Table 1 and Figure 2 show that COMMAND consistently outperforms all baselines across the evaluated datasets. This performance gain is due to two key factors. First, recruiting multiple agents expands the solution set and increases answer diversity, giving the principal a richer pool to choose from and a higher chance of including a high-quality reasoning path than few-shot CoT or the principal acting alone. Second, COMMAND connects agent utility to the principal's ranking feedback, as agents are rewarded for producing answers that score well under the principal's evaluation, which steers search toward faithful, well-justified solutions and induces productive competition rather than redundant exploration.

**COMMAND Enables Competition Among Heterogeneous Agents.** Despite substantial differences in dataset difficulty and baseline accuracy, Table 2 shows that most agents benefit from COMMAND. As expected, COMMAND induces competition among agents for receiving higher utilities. The ranking-based utility induces competition and a winner-takes-more effect: an initially stronger agent gains the most, while weaker agents might be mislead. For example, stronger agents with higher accuracy such as Mistral on Math and Phi on GSM8K show some gains after competition. By contrast, the weakest agents that initially have the lowest accuracy exhibit small declines such as Falcon on Math and Zephyr on GSM-Hard.

Table 1: Accuracy (%) across benchmark datasets for each method. **Bold** indicates the best performance for each dataset.

| Dataset | Few-shot CoT | Principal | rStar | COMMAND |
|---|---|---|---|---|
| Math | 8.0 | 4.8 | 26.8 | **29.1** |
| GSM8K | 15.7 | 35.8 | 50.0 | **60.3** |
| GSM-Hard | 20.6 | 11.7 | 28.1 | **29.4** |

## 4.3 VALIDATION FOR ASSUMPTION 1

We empirically verify Assumption 1 in the experiments. Given the candidate answers generated by all the agents, we estimate the principal's utility by submitting all agent's answers to the principal,

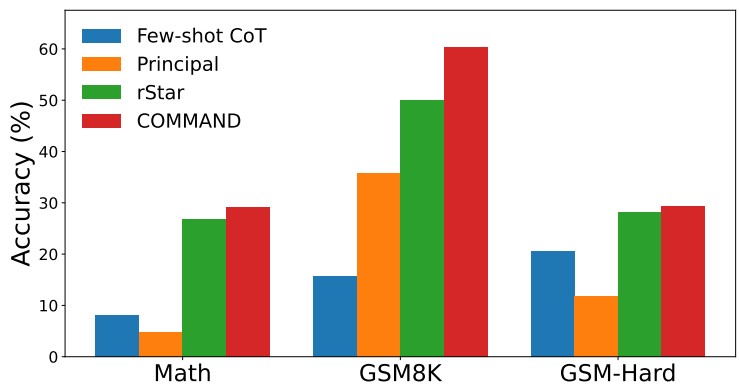

Figure 2: Accuracy comparison across benchmark datasets for each method.

Table 2: Accuracy (%) of each model before and after applying COMMAND across three benchmark datasets. **Bold** highlights the best result for each dataset.

| Agent | Math | | Agent | GSM8K | | GSM-Hard | |
|---|---|---|---|---|---|---|---|
| | **Before** | **After** | | **Before** | **After** | **Before** | **After** |
| Mistral | 31.4 | **33.1** | Mistral | 39.0 | 39.3 | 27.2 | 27.5 |
| Zephyr | 28.4 | 28.8 | Zephyr | 60.7 | 61.0 | 22.2 | 21.9 |
| Falcon | 18.7 | 16.1 | Phi | 58.7 | **61.7** | 30.3 | **31.2** |

collecting the feedback, and normalizing the scores. For part (i), we denote a problem valid if no alternative answer yields strictly higher utility for both the principal and the agent than the submitted answers across all iterations. We then report the percentage of valid problems for each dataset and agent. As shown in Table 3, this percentage is consistently around 90% across agents and datasets, meaning that most of the problems satisfy the Pareto-optimal play in Assumption 1. For part (iii), we compute the Pearson correlation between the principal's utilities and those of each agent for each dataset. Table 4 reports the percentage of positively correlated problems and average correlation coefficients, which are positive for all datasets and agents. This result justifies the non-negative alignment hypothesis in Assumption 1. Although the principal's and agent's objectives are not perfectly aligned, solving the same task induces partial alignment, so utilities move together on average rather than in opposite directions.

Table 3: Percentage of valid problems across all agents and datasets.

| Agent | Math | Agent | GSM8K | GSM-Hard |
|---|---|---|---|---|
| Mistral | 90.7 | Mistral | 91.0 | 97.6 |
| Zephyr | 90.7 | Zephyr | 90.6 | 95.5 |
| Falcon | 87.6 | Phi | 91.4 | 96.0 |

Table 4: Verification of non-negative alignment between principal and agents. *Positive* (%) is the percentage of problems with positive correlation. *Average* is the mean correlation across problems.

| Agent | Math | | Agent | GSM8K | | GSM-Hard | |
|---|---|---|---|---|---|---|---|
| | **Positive** | **Average** | | **Positive** | **Average** | **Positive** | **Average** |
| Mistral | 91.3 | 0.4934 | Mistral | 72.9 | 0.2611 | 82.7 | 0.4063 |
| Zephyr | 93.6 | 0.5026 | Zephyr | 72.6 | 0.2672 | 82.9 | 0.3820 |
| Falcon | 92.0 | 0.4761 | Phi | 66.2 | 0.1561 | 72.4 | 0.2204 |

## 5 RELATED WORKS

**LLM Diverse Reasoning.** A growing body of work improves LLM performance at inference time, often referred to as *test-time compute*. Popular approaches include (i) prompt-based techniques, such as chain-of-thought prompting to elicit structured reasoning (Wei et al., 2022), and (ii) sampling and search, including Top-$k$, Top-$p$, beam search (Feng et al., 2023), or tree-based exploration such as MCTS (Sutton et al., 1998). To further refine candidate outputs, methods such as majority voting (Wang et al., 2022) and verifier models (Lightman et al., 2024) have been employed to select high-quality responses. A key insight is that sampling diverse reasoning paths (either entire trajectories or step-by-step expansions) significantly outperforms simply picking the most probable answers without exploration, in terms of accuracy and task completion rates (Snell et al., 2024; Brown et al., 2024). Our method builds on this paradigm by framing inference as a game-theoretic process: multiple agents strategically generate diverse answers guided by designed prompts. Through reward design and competition in a repeated setting, the system incentivizes exploration of higher-quality reasoning paths while maintaining diversity.

**LLM Self-improvement.** A growing line of work explores how LLMs improve reasoning through structured self-improvement without external supervision. Inspired by AlphaZero where learning emerges from play and feedback (Silver et al., 2017), LLMs can iteratively provide feedback, refinements, or critiques to improve its answers after generations (Madaan et al., 2023; Cheng et al., 2024; Chen et al., 2024). However, the effectiveness of this process often depends on the model's inherent capabilities and may yield misleading gains in weaker models. Our approach aims to enable LLMs to self-improve by learning from feedback provided by a principal without relying on RL or fine-tuning.

**LLMs as Strategic Agents.** With the advancement of LLMs, a growing body of research investigates their behavior in game-theoretic multi-agent settings. Empirical studies and benchmarks analyze LLM decision-making across diverse games, either collaborative or adversarial, with two or multiple agents, and under short- or long-term utility objectives (Lan et al., 2023; Huang et al., 2024a; Piatti et al., 2024). Beyond evaluation, multiple LLMs have been organized into multi-agent systems that engage in debate, feedback exchange, or negotiation to improve answer quality (Huang et al., 2024b; Chen et al., 2025; Wang et al., 2024a). Current work designs game-theoretic frameworks that directly enhance LLM reasoning and consistency. For example, adversarial games assign LLMs to attacker and defender roles (Cheng et al., 2024; Kirchner et al., 2024), though these typically require reinforcement learning to train the policies and are often limited to two-agent settings. The consensus game framework in Jacob et al. (2023) offers a training-free approach that aligns generation and discrimination to promote consistency, but it also remains constrained to two agents and may converge to suboptimal equilibria. Our method extends this line of work by modeling multi-agent interactions with structured feedback to promote diverse reasoning and strategic improvement without fine-tuning or reinforcement learning.

## 6 CONCLUSIONS

In this work, we presented COMMAND, a competitive multi-agent framework for improving the reasoning capabilities of LLMs. By framing the interaction between a principal and multiple agents as a delegation process, COMMAND leverages competition and alignment incentives to elicit higher-quality answers without additional training. We established theoretical guarantees showing that multi-agent systems provably outperform single-agent setups under fair comparisons. Furthermore, using online mirror descent, each agent achieves sublinear regret, ensuring that its average performance approaches that of the best fixed policy. The agents' average policies converge to a Nash equilibrium, aligning with the principal's feedback while preserving their own reasoning-based preferences. Empirical evaluations across diverse mathematical reasoning benchmarks further demonstrated consistent improvements in factual accuracy.

While effective, current experiments focus on math reasoning, so future work includes extending experiments to other reasoning-centric tasks such as long-form question answering, multi-step planning, and theorem-style proofs. Furthermore, incorporating additional tools from game theory and mechanism design further enhance alignment and robustness among LLM agents, especially in open-ended or adversarial settings.

## REPRODUCIBILITY STATEMENT

We have taken several steps to facilitate reproducibility. The experimental setup and datasets are described in Section 4.1. Details of computational resources are reported in Appendix B. All proofs and explanations of assumptions are provided in Appendix A. An anonymized code repository to reproduce our results can be assessed via `https://anonymous.4open.science/r/ICLR2026_COMMAND_algorithm-DCAC`.

## THE USE OF LARGE LANGUAGE MODELS (LLMS)

In preparing this manuscript, we employed LLMs, specifically ChatGPT-4o and ChatGPT-5, as writing assistants. Their use was limited to polishing grammar, improving fluency, correcting LaTeX code (e.g., tables and formatting), and assisting with code suggestions and debugging. All substantive ideas, analyses, and conclusions remain the original contributions of the authors.

## ETHICS STATEMENT

We adhere to the ICLR Code of Ethics in all aspects of this work. Our experiments use licensed public datasets and open-source models; we document preprocessing and release code, prompts, and evaluation details to support reproducibility. The methods are not intended for high-risk uses; any deployment should include domain-specific safeguards and oversight.

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

# A  PROOFS

## A.1  PROOF OF THEOREM 1

**Lemma 1** (Kleinberg & Kleinberg (2018)). *If $M$ is any mechanism and $\sigma$ is a best–response strategy (profile) to $M$. Let $f_{M,\sigma}$ denote the interim allocation function, i.e., the function that specifies an outcome of $M$ when agents follow $\sigma$. Then there exists a single proposal mechanism $M'$ and a best response $\sigma'$ to $M'$, such that the interim allocation functions $f_{M,\sigma}$ and $f_{M',\sigma'}$ are identical.*

*Proof of Theorem 1 (part a).* By Lemma 1, it suffices to consider the case when $M$ is a single proposal mechanism. Let the actual answers in the single proposal mechanism $P$ be

$$\bar\omega = \{\omega_1, \omega_2, \ldots, \omega_k\}.$$

Consider a partition of $\bar\omega$ as follows

$$\bar\omega_1 = \{\omega_1, \ldots, \omega_{k_1}\},$$

$$\bar\omega_2 = \{\omega_{k_1+1}, \ldots, \omega_{k_1+k_2}\},$$

$$\vdots$$

$$\bar\omega_N = \{\omega_{k_1+\ldots+k_{N-1}+1}, \ldots, \omega_{k_1+\ldots+k_N=k}\}.$$

Suppose that agent $i$'s answers in $P'$ is $\bar\omega_i$ so that the outcomes of $P$ and $P'$ are coupled. This coupling is indeed possible because the probability that agent $i$'s answers are within $\bar\omega_i = \{\omega_{i_1}, \ldots, \omega_{i_{k_i}}\}$ is exactly equal to the marginal probability that $\bar\omega$'s $i$-th partition is $\bar\omega_i = \{\omega_{i_1}, \ldots, \omega_{i_{k_i}}\}$. Consider a single proposal mechanism $P$ and corresponding equilibrium strategies $\sigma$, principal's utility function $U$ and agents's utility function $U_y$. Given the answers above, single proposal mechanism will select the answer which maximizes $U_y(\omega_j)$ for $j \in [k]$, i.e.,

$$\omega_1^* = \arg\max_{j\in[N]} U_y(\omega_j).$$

Let this answer be $\omega_1^*$. Now consider a multi-agent single proposal mechanism $P'$, and suppose that the agents play equilibrium strategies $\sigma'$. Let $\omega_2^*$ be the corresponding winner in this case.

We want to show that $U(\omega_2^*) \geq U(\omega_1^*)$ for any answers $\bar\omega$. Suppose not. If $\omega_1^*$ and $\omega_2^*$ belong to the same partition $\bar\omega_i$, it means that agent $i$ has both the answers $\omega_1^*$ and $\omega_2^*$. By our definition on $\omega_1^*$, submitting $\omega_1^*$ gives the better or the same utility for agent $i$. If $U_y(\omega_1^*) > U_y(\omega_2^*)$, then simply submitting $\omega_1^*$ gives a strictly better utility for agent $i$, and it contradicts that $\sigma'$ is an equilibrium strategy. If $U_y(\omega_1^*) = U_y(\omega_2^*)$, then $U(\omega_1^*) \leq U(\omega_2^*)$ by Assumption 1. Hence, $U(\omega_1^*) \leq U(\omega_2^*)$ in both cases.

Now suppose that $\omega_1^*$ and $\omega_2^*$ belong to the different partitions $\bar\omega_i$ and $\bar\omega_j$. Suppose that agent $i$'s equilibrium strategy satisfies $\sigma_i' \neq \omega_1^*$. If remaining $\bar\omega_i$ is strictly better for agent $i$, it contradicts that $\sigma_i'$ is an equilibrium strategy. Hence $U_y(\omega_1^*) = U_y(\sigma_i')$. Again by Assumption 1, since agent $i$ plays $\sigma_i'$, we have $U(\sigma_i') \geq U(\omega_1^*)$. Note that the principal should observe both $\sigma_i'$ and $\omega_2^*$, and commits to $\omega_2^*$. This implies that $U(\omega_2^*) \geq U(\sigma_i') \geq U(\omega_1^*)$, and we finish the proof. $\square$

For part b, We begin by defining a key event in which the answer chosen by an agent to maximize their own expected utility is also the one most preferred by the principal. Under the condition of Theorem 1, the agent's expected utility for proposing $\omega$ is approximately

$$[\frac{U(\omega)+1}{2}]^{k(N-1)} U_y(\omega),$$

where $k(N-1)$ is the total number of competing candidate answers from the other agents. This motivates the agent to select

$$\arg\max_{\omega\in\bar\omega_i}\{[U(\omega)+1]/2\}^{k(N-1)} U_y(\omega).$$

However, the system designer would ideally prefer the agent to select $\arg\max_{\omega\in\bar{\omega}_i} U(\omega)$, which aligns with the principal's interest. We therefore define the event $E_i$ under which the agent's optimal choice (based on their own objective) coincides with the principal's preferred one:

$$E_i(\{U(\omega), U_y(\omega)\}_{\omega\in\bar{\omega}_i}) = \left\{\bar{\omega}_i : \arg\max_{\omega\in\bar{\omega}_i}\{[U(\omega)+1]/2\}^{k(N-1)}U_y(\omega) = \arg\max_{\omega\in\bar{\omega}_i} U(\omega)\right\}.$$

In other words, this event captures the favorable case when the agent's self-interested decision also maximizes the principal's utility. For notational simplicity, we refer to $E_i(\{U(\omega), U_y(\omega)\}_{\omega\in\bar{\omega}_i})$ simply as $E_i$ or $E_i(\bar{\omega}_i)$. At the joint level, define the event:

$$E(\bar{\omega}) := E(U(\bar{\omega}), U_y(\bar{\omega})) = \bigcap_{i\in[N]} E_i(\bar{\omega}_i),$$

which holds if all agents simultaneously select answers that align with the principal's preference.

Now, we will prove that given $E(U(\bar{\omega}), U_y(\bar{\omega}))$ and the other's strategies, proposing a answer that maximizes $U(\cdot)$ will be approximately best-response.

Let $\sigma_i^x$ be a strategy to propose a answer that maximizes $U(\cdot)$ for agent $i$. Given that all the other agents $j$ playing $\sigma_j^x$ for $j\neq i$, we can further obtain the following regarding agent $i$'s utility:

$$\begin{aligned}
U_i(\sigma_i^x, \sigma_{-i}) &= \mathbb{E}\left[U_i(\sigma_i^x, \sigma_{-i}) \mid E\right]\mathbb{P}[E] + \mathbb{E}\left[U_i(\sigma_i^x, \sigma_{-i}) \mid E^c\right]\mathbb{P}[E^c] \\
&\geq \mathbb{E}\left[U_i(\sigma_i^x, \sigma_{-i}) \mid E\right]\mathbb{P}[E] - \mathbb{P}[E^c] \\
&\geq \mathbb{E}\left[U_i(\sigma_i', \sigma_{-i}) \mid E\right]\mathbb{P}[E] - \mathbb{P}[E^c] \\
&= \mathbb{E}\left[U_i(\sigma_i', \sigma_{-i})\right] - \mathbb{E}\left[U_i(\sigma_i', \sigma_{-i} \mid E^c)\right]\mathbb{P}[E^c] - \mathbb{P}[E^c] \\
&\geq \mathbb{E}\left[U_i(\sigma_i', \sigma_{-i})\right] - 2\mathbb{P}[E^c],
\end{aligned}$$

where the second inequality follows from the fact that given $E$, playing $\sigma_i^x$ is weakly dominant over any other strategy $\sigma_i'$ for agent $i$.

This implies that if we characterize a good lower bound $\alpha$ such that $\mathbb{P}[E] \geq \alpha$, we have

$$U_i(\sigma_i^x, \sigma_{-i}) \geq U_i(\sigma_i', \sigma_{-i}) - 2(1-\alpha),$$

which implies that $\sigma_i^x$ is $2(1-\alpha)$-approximate BNE. The next step will be to consider the lower bound of event $E$.

**Lemma 2** (Shin et al. (2023)). *Under Assumption 1, assume the utility function of principals and agents are independent, we have $\mathbb{P}[E] \geq \alpha_0 = e^{-\frac{N^2}{2(N-1)^2}}$.*

**Lemma 3.** *Suppose $U(\omega)$ and $U_y(\omega)$ are positively correlated. Then the probability of the event $E_i(\bar{\omega}_i)$ is weakly greater than in the case where $U(\omega)$ and $U_y(\omega)$ are independent:*

$$\mathbb{P}_{U\perp U_y}[E_i(\bar{\omega}_i)] \leq \mathbb{P}_{U\uparrow U_y}[E_i(\bar{\omega}_i)].$$

*Proof.* We prove the lemma using a coupling argument based on the stochastic dominance induced by positive correlation.

Let $\bar{\omega}_i$ be the set of proposals for agent $i$, with $|\bar{\omega}_i| = k$. Let $\omega^* = \operatorname{argmax}_{\omega\in\bar{\omega}_i} U(\omega)$ be the proposal that maximizes $U$, and let $x_1 = U(\omega^*)$. For $j = 2, \ldots, k$, let $x_j$ be the $x$-values of the other proposals, ordered such that $x_1 \geq x_2 \geq \cdots \geq x_k$. Define $B_j = \left(\frac{x_j}{x_1}\right)^{k(N-1)}$ for $j = 2, \ldots, k$.

The event $E_i$ occurs if and only if for all $j = 2, \ldots, k$,

$$\frac{U_y(\omega^*)}{U_y(\omega_j)} \geq B_j.$$

Define $C_j = \left\{\frac{U_y(\omega^*)}{U_y(\omega_j)} \geq B_j\right\}$, so that $E_i = \cap_{j=2}^k C_j$.

Now, condition on the $x$-values $\boldsymbol{x} = (x_1, \ldots, x_k)$. In the independent case, $U_y(\omega)$ is drawn from the marginal distribution $F_y$ for each $\omega$, independently. In the positively correlated case, $U_y(\omega)$ is drawn

from the conditional distribution $F_{y|x(\omega)}$. By positive correlation, for any $x_1 > x_j$, the distribution $F_{y|x_1}$ stochastically dominates $F_{y|x_j}$, i.e.,

$$\mathbb{P}[y > t \mid x_1] \geq \mathbb{P}[y > t \mid x_j] \quad \forall t.$$

In particular, $F_{y|x_1}$ stochastically dominates the marginal distribution $F_y$, and $F_y$ stochastically dominates $F_{y|x_j}$ for $j \geq 2$.

By stochastic dominance, we can construct coupled random variables as follows:

- For $U_y(\omega^*)$, let $Y_1^+ \sim F_{y|x_1}$ and $Y_1^* \sim F_y$ such that $Y_1^+ \geq Y_1^*$ almost surely.

- For each $U_y(\omega_j)$ with $j \geq 2$, let $Y_j^- \sim F_{y|x_j}$ and $Y_j^* \sim F_y$ such that $Y_j^- \leq Y_j^*$ almost surely.

Such couplings exist due to the stochastic dominance relations.

Now, for each $j \geq 2$, almost surely,

$$\frac{Y_1^+}{Y_j^-} \geq \frac{Y_1^*}{Y_j^*}.$$

Therefore,

$$\left\{ \frac{Y_1^*}{Y_j^*} \geq B_j \right\} \subseteq \left\{ \frac{Y_1^+}{Y_j^-} \geq B_j \right\}.$$

This implies that for each $j$,

$$\mathbb{P}\left[ \frac{Y_1^+}{Y_j^-} \geq B_j \right] \geq \mathbb{P}\left[ \frac{Y_1^*}{Y_j^*} \geq B_j \right].$$

Since the $y$-values are conditionally independent given $\boldsymbol{x}$, the events $C_j$ are conditionally independent given $y(\omega^*)$ in both cases. However, by the above coupling, we have almost surely,

$$\bigcap_{j=2}^{k} \left\{ \frac{Y_1^*}{Y_j^*} \geq B_j \right\} \subseteq \bigcap_{j=2}^{k} \left\{ \frac{Y_1^+}{Y_j^-} \geq B_j \right\}.$$

Thus, the joint probability satisfies:

$$\mathbb{P}\left[ \bigcap_{j=2}^{k} \left\{ \frac{Y_1^+}{Y_j^-} \geq B_j \right\} \mid \boldsymbol{x} \right] \geq \mathbb{P}\left[ \bigcap_{j=2}^{k} \left\{ \frac{Y_1^*}{Y_j^*} \geq B_j \right\} \mid \boldsymbol{x} \right].$$

The right-hand side is the conditional probability of $E_i$ in the independent case, and the left-hand side is the conditional probability in the positively correlated case.

Taking expectation over $\boldsymbol{x}$, we obtain:

$$\mathbb{P}_{U \uparrow U_y}[E_i(\bar{\omega}_i)] \geq \mathbb{P}_{U \perp U_y}[E_i(\bar{\omega}_i)],$$

which completes the proof. $\qquad\square$

By Lemma 2, when $U(\omega)$ and $U_y(\omega)$ are independent and identically distributed across candidates, the probability of the alignment event $E$ admits an explicit lower bound. This implies the existence of a $2(1 - \alpha)$-approximate BNE under the independent setting, where $\alpha = 1 - \mathbb{P}[E]$. By Lemma 3, when $U(\omega)$ and $U_y(\omega)$ are positively correlated, the probability of $E$ is weakly greater than in the independent case. Therefore, the same approximation bound holds in the positively correlated setting.

## A.2  Proof of Theorem 2

We begin with some auxiliary lemmas that characterize the structure of the entropy-regularized policy updates. Specifically, we leverage the optimality conditions induced by mirror descent with negative entropy to relate policy differences to KL divergence terms. These lemmas follow standard arguments in the online learning literature, but we include them here for completeness and to establish notation for our subsequent regret analysis (Jacob et al., 2022; Bakhtin et al., 2022; Shalev-Shwartz, 2012).

**Lemma 4.** *At any round $t$, if player $i$'s policy update follows:*

$$\pi_i^{t+1} = \arg\max_{\pi} \left\{ \sum_{t'=1}^{t} U_i^{t'}(\pi) - \frac{1}{\eta} \varphi(\pi) \right\},$$

*where the regularizer is the negative entropy,*

$$\varphi(\pi) := \sum_{a \in \mathcal{A}_i} \pi(a) \log \pi(a),$$

*then the resulting policy $\pi_i^{t+1}$ is equivalent to the one generated by Algorithm 1.*

*Proof.* We begin by rewriting the cumulative utility term as a linear function over actions:

$$\sum_{t'=1}^{t} U_i^{t'}(\pi) = \sum_{a \in \mathcal{A}_i} \left( \sum_{t'=1}^{t} U_i(a, a_{-i}^{t'}) \right) \pi(a).$$

Substituting into the objective, the optimization becomes:

$$\max_{\pi} \left\{ \sum_{a \in \mathcal{A}_i} \left( \sum_{t'=1}^{t} U_i(a, a_{-i}^{t'}) \right) \pi(a) - \frac{1}{\eta} \sum_{a \in \mathcal{A}_i} \pi(a) \log \pi(a) \right\}.$$

This is a standard instance of entropy-regularized maximization over a simplex. Its answer is well known to be a softmax distribution over accumulated utilities:

$$\pi_i^{t+1}(a) = \frac{\exp\left( \eta \sum_{t'=1}^{t} U_i(a, a_{-i}^{t'}) \right)}{\sum_{a' \in A_i} \exp\left( \eta \sum_{t'=1}^{t} U_i(a', a_{-i}^{t'}) \right)}.$$

This precisely matches the update rule employed in Algorithm 1, where action values are incrementally aggregated and exponentiated with temperature $\eta$. Thus, the two procedures are equivalent. □

**Lemma 5.** *Suppose that at each round $t + 1$, player $i$ updates their policy via the following optimization:*

$$\pi_i^{t+1} = \arg\max_{\pi} \left\{ \sum_{t'=1}^{t} \tilde{U}_i(\pi, a_{-i}^{t'}) - \frac{1}{\eta} \varphi(\pi) \right\},$$

*where $\eta > 0$ is a fixed parameter, the regularizer $\varphi$ is the negative Shannon entropy*

$$\varphi(\pi) := \sum_{a \in \mathcal{A}_i} \pi(a) \log \pi(a),$$

*and the shifted utility function is defined as*

$$\tilde{U}_i(a, a_{-i}^t) := U_i(a, a_{-i}^t) - \min_{\mathbf{a} \in \mathcal{A}_1 \times \cdots \times \mathcal{A}_N} U_i(\mathbf{a}).$$

*Then, the resulting policy admits a softmax representation:*

$$\pi_i^{t+1}(a) = \frac{\exp\left( v_i^{t+1}(a) \right)}{\sum_{a' \in \mathcal{A}_i} \exp\left( v_i^{t+1}(a') \right)} \quad \forall a \in \mathcal{A}_i,$$

*where*

$$v_i^{t+1}(a) := \eta \sum_{t'=1}^{t} \tilde{U}_i(a, a_{-i}^{t'}).$$

*Proof.* To simplify notation, let $\gamma := \min_{\mathbf{a} \in \mathcal{A}_1 \times \cdots \times \mathcal{A}_N} \tilde{U}_i(\mathbf{a})$. Since the minimum utility is constant across all actions, subtracting it from the original utilities does not affect the maximizer of the objective. Thus, the cumulative utility component can be rewritten as:

$$\sum_{t'=1}^{t} \tilde{U}_i(\pi, a_{-i}^{t'}) = \sum_{a \in \mathcal{A}_i} \left( \sum_{t'=1}^{t} \tilde{U}_i(a, a_{-i}^{t'}) \right) \pi(a).$$

Substituting into the objective function yields:

$$\pi_i^{t+1} = \arg\max_{\pi} \left\{ \eta \sum_{a \in \mathcal{A}_i} \left( \sum_{t'=1}^{t} \tilde{U}_i(a, a_{-i}^{t'}) \right) \pi(a) - \sum_{a \in \mathcal{A}_i} \pi(a) \log \pi(a) \right\}.$$

This is a classical instance of entropy-regularized linear optimization over the probability simplex. The optimal answer is known to be a softmax distribution over the accumulated (shifted) utility scores:

$$\pi_i^{t+1}(a) = \frac{\exp\left(v_i^{t+1}(a)\right)}{\sum_{a' \in A_i} \exp\left(v_i^{t+1}(a')\right)} \quad \forall a \in \mathcal{A}_i,$$

where $v_i^{t+1}(a) := \eta \sum_{t'=1}^{t} \tilde{U}_i(a, a_{-i}^{t'})$, completing the proof. $\square$

**Lemma 6.** *Let $t \geq 1$ and fix agent $i$. Suppose $\pi_i^t$ and $\pi_i^{t+1}$ are the policy updates produced by Algorithm 1 at iteration $t$ and $t+1$ respectively. Then for any pair of distributions $\pi, \pi'$, the following identity holds:*

$$\left\langle -\eta U_i^t + \nabla\varphi(\pi_i^{t+1}) - \nabla\varphi(\pi_i^t), \pi - \pi' \right\rangle = 0.$$

*Proof.* We analyze the policy update dynamics via the optimality conditions associated with the mirror descent updates under negative entropy regularization. Define the empirical utility vectors at rounds $t-1$ and $t$ as

$$\bar{U}_i^{t-1} := \frac{1}{t-1} \sum_{s=1}^{t-1} u_i^s, \qquad \bar{U}_i^t := \frac{1}{t} \sum_{s=1}^{t} u_i^s.$$

From the KKT conditions of the optimization problems defining $\pi_i^t$ and $\pi_i^{t+1}$, we know that:

$$\left\langle -\bar{U}_i^t + \frac{1}{\eta t} \nabla\varphi(\pi_i^{t+1}), \pi - \pi' \right\rangle = 0, \quad \left\langle -\bar{U}_i^{t-1} + \frac{1}{\eta(t-1)} \nabla\varphi(\pi_i^t), \pi - \pi' \right\rangle = 0.$$

Subtracting the second equation from the first gives:

$$0 = \left\langle \bar{U}_i^{t-1} - \bar{U}_i^t + \frac{1}{\eta t} \nabla\varphi(\pi_i^{t+1}) - \frac{1}{\eta(t-1)} \nabla\varphi(\pi_i^t), \ \pi - \pi' \right\rangle.$$

Next, observe that:

$$\bar{U}_i^t = \frac{t-1}{t} \bar{U}_i^{t-1} + \frac{1}{t} U_i^t \quad \Rightarrow \quad \bar{U}_i^t - \bar{U}_i^{t-1} = -\frac{1}{t} \bar{U}_i^{t-1} + \frac{1}{t} U_i^t, \quad \text{or} \quad \bar{U}_i^{t-1} - \bar{U}_i^t = \frac{1}{t}(\bar{U}_i^{t-1} - U_i^t).$$

Substituting this into the previous expression yields:

$$0 = \left\langle \frac{1}{t}(\bar{U}_i^{t-1} - u_i^t) + \frac{1}{\eta t} \nabla\varphi(\pi_i^{t+1}) - \frac{1}{\eta(t-1)} \nabla\varphi(\pi_i^t), \ \pi - \pi' \right\rangle.$$

Rewriting $\bar{U}_i^{t-1} = \bar{U}_i^t - \frac{1}{t}(u_i^t - \bar{U}_i^t)$ and simplifying, we reach:

$$0 = \left\langle \frac{-1}{t-1} U_i^t + \frac{1}{\eta(t-1)} \nabla\varphi(\pi_i^{t+1}) - \frac{1}{\eta(t-1)} \nabla\varphi(\pi_i^t), \ \pi - \pi' \right\rangle.$$

Multiplying through by $\eta(t-1)$ gives the desired result:

$$\left\langle -\eta U_i^t + \nabla\varphi(\pi_i^{t+1}) - \nabla\varphi(\pi_i^t), \pi - \pi' \right\rangle = 0.$$

$\square$

**Lemma 7.** *Let $i$ be any agent and $t \geq 1$. Suppose that the policies $\pi_i^t$ and $\pi_i^{t+1}$ are the successive updates obtained from Algorithm 1. Then, for any policy $\pi$, the following identity holds:*

$$\left\langle -U_i^t, \ \pi - \pi_i^{t+1} \right\rangle = \frac{1}{\eta} \left( D_{\mathrm{KL}}(\pi \| \pi_i^t) - D_{\mathrm{KL}}(\pi \| \pi_i^{t+1}) + D_{\mathrm{KL}}(\pi_i^{t+1} \| \pi_i^t) \right).$$

*Proof.* We begin by recalling from Lemma 6 that the following condition is satisfied by the update rule for any pair $\pi, \pi'$:

$$\left\langle -\eta U_i^t + \nabla\varphi(\pi_i^{t+1}) - \nabla\varphi(\pi_i^t), \ \pi - \pi' \right\rangle = 0.$$

By choosing $\pi' = \pi_i^{t+1}$, this reduces to:

$$\left\langle -\eta U_i^t + \nabla\varphi(\pi_i^{t+1}) - \nabla\varphi(\pi_i^t), \ \pi - \pi_i^{t+1} \right\rangle = 0.$$

Rearranging terms gives:

$$\eta \left\langle -U_i^t, \pi - \pi_i^{t+1} \right\rangle = \left\langle \nabla\varphi(\pi_i^t) - \nabla\varphi(\pi_i^{t+1}), \ \pi - \pi_i^{t+1} \right\rangle.$$

We now invoke the identity for the Bregman divergence associated with the negative entropy function $\varphi$, which yields:

$$\left\langle \nabla\varphi(\pi_i^t) - \nabla\varphi(\pi_i^{t+1}), \ \pi - \pi_i^{t+1} \right\rangle = D_{\mathrm{KL}}(\pi \| \pi_i^t) - D_{\mathrm{KL}}(\pi \| \pi_i^{t+1}) + D_{\mathrm{KL}}(\pi_i^{t+1} \| \pi_i^t).$$

Putting everything together, we obtain:

$$\eta \left\langle -U_i^t, \pi - \pi_i^{t+1} \right\rangle = D_{\mathrm{KL}}(\pi \| \pi_i^t) - D_{\mathrm{KL}}(\pi \| \pi_i^{t+1}) + D_{\mathrm{KL}}(\pi_i^{t+1} \| \pi_i^t).$$

Dividing both sides by $\eta$ completes the proof. $\qquad\square$

**Lemma 8.** *For any agent $i$ and round $t$, the following upper bound holds for all policies $\pi$:*

$$U_i^t(\pi) - U_i^t(\pi_i^t) \leq \frac{\eta \|U_i^t\|_\infty^2}{4} - \frac{1}{\eta} D_{\mathrm{KL}}(\pi \| \pi_i^{t+1}) + \frac{1}{\eta} D_{\mathrm{KL}}(\pi \| \pi_i^t).$$

*Proof.* We begin by illustrating Lemma 6, which characterizes the optimality condition at each step via:

$$\left\langle -\eta U_i^t + \nabla\varphi(\pi_i^{t+1}) - \nabla\varphi(\pi_i^t), \ \pi - \pi_i^{t+1} \right\rangle = 0 \quad \text{for all } \pi.$$

Rearranging gives:

$$\left\langle U_i^t, \pi - \pi_i^{t+1} \right\rangle = \frac{1}{\eta} \left\langle \nabla\varphi(\pi_i^{t+1}) - \nabla\varphi(\pi_i^t), \ \pi - \pi_i^{t+1} \right\rangle.$$

We now apply the three-point identity for Bregman divergence induced by the negative entropy regularizer:

$$\left\langle \nabla\varphi(\pi_i^{t+1}) - \nabla\varphi(\pi_i^t), \ \pi - \pi_i^{t+1} \right\rangle = D_{\mathrm{KL}}(\pi \| \pi_i^t) - D_{\mathrm{KL}}(\pi \| \pi_i^{t+1}) + D_{\mathrm{KL}}(\pi_i^{t+1} \| \pi_i^t).$$

Substituting this gives:

$$\left\langle U_i^t, \ \pi - \pi_i^{t+1} \right\rangle = \frac{1}{\eta} \left( D_{\mathrm{KL}}(\pi \| \pi_i^t) - D_{\mathrm{KL}}(\pi \| \pi_i^{t+1}) + D_{\mathrm{KL}}(\pi_i^{t+1} \| \pi_i^t) \right).$$

To relate this to $U_i^t(\pi) - U_i^t(\pi_i^t)$, we subtract and add $U_i^t(\pi_i^t)$, and observe:

$$U_i^t(\pi) - U_i^t(\pi_i^t) = \left\langle U_i^t, \ \pi - \pi_i^{t+1} \right\rangle + \left\langle U_i^t, \ \pi_i^{t+1} - \pi_i^t \right\rangle.$$

Combining with the expression above, we have:

$$U_i^t(\pi) - U_i^t(\pi_i^t) = \frac{1}{\eta} \left( D_{\mathrm{KL}}(\pi \| \pi_i^t) - D_{\mathrm{KL}}(\pi \| \pi_i^{t+1}) + D_{\mathrm{KL}}(\pi_i^{t+1} \| \pi_i^t) \right) + \left\langle U_i^t, \ \pi_i^{t+1} - \pi_i^t \right\rangle.$$

Next, apply Young's inequality to the final term:

$$\left\langle U_i^t, \pi_i^{t+1} - \pi_i^t \right\rangle \leq \frac{\eta}{4} \|U_i^t\|_\infty^2 + \frac{1}{\eta} \|\pi_i^{t+1} - \pi_i^t\|_1^2.$$

Finally, use the strong convexity of KL divergence to bound:

$$\|\pi_i^{t+1} - \pi_i^t\|_1^2 \le 2D_{\mathrm{KL}}(\pi_i^{t+1}\|\pi_i^t),$$

and thus:

$$\langle U_i^t,\ \pi_i^{t+1} - \pi_i^t \rangle \le \frac{\eta}{4}\|U_i^t\|_\infty^2 + \frac{2}{\eta}D_{\mathrm{KL}}(\pi_i^{t+1}\|\pi_i^t).$$

Putting all terms together, we arrive at:

$$U_i^t(\pi) - U_i^t(\pi_i^t) \le \frac{\eta}{4}\|U_i^t\|_\infty^2 + \frac{1}{\eta}D_{\mathrm{KL}}(\pi\|\pi_i^t) - \frac{1}{\eta}D_{\mathrm{KL}}(\pi\|\pi_i^{t+1}),$$

as desired. $\qquad\square$

*Proof of Theorem 2.* Summing Lemma 8 over $t = 0$ to $T$, the KL divergence terms telescope. Since $D_{\mathrm{KL}}(\pi\|\pi_i^{T+1}) \ge 0$, we have:

$$\sum_{t=1}^T U_i^t(\pi) - U_i^t(\pi_i^t) \le \frac{\eta}{4}\sum_{t=1}^T \|U_i^t\|_\infty^2 + \frac{1}{\eta}D_{\mathrm{KL}}(\pi\|\pi_i^0)$$

$$\le \frac{\eta T}{4} + \frac{D_{\mathrm{KL}}(\pi\|\pi_i^0)}{\eta},$$

where the last inequality uses $\|U_i^t\|_\infty \le 1$. Setting $\eta = \frac{1}{\sqrt{T}}$ completes the proof. $\qquad\square$

### A.3 PROOF OF THEOREM 3

In this section, we present the proof of Theorem 3.

*Proof of Theorem 3.* Fix an agent $i \in [N]$, and any policy $\pi^*$, and introduce the discrete-time stochastic process

$$w^t := \big(U_i(\pi^*, \pi_{-i}^t) - U_i(\pi_i^t, \pi_{-i}^t)\big) - \big(U_i(\pi^*, a_{-i}^t) - U_i(\pi_i^t, a_{-i}^t)\big).$$

Since each opponent player $j \ne i$ plays according to Algorithm 1, the answers $a_{-i}^t$ at each round $t$ is sampled from the joint policy $\pi_{-i}^t$. Therefore, $w^t$ is a martingale difference sequence. Furthermore, by expanding the definition of $U_i$, the absolute value of $w^t$ satisfies

$$\begin{aligned}
|w^t| &= \big|\big(U_i(\pi^*, \pi_{-i}^t) - U_i(\pi_i^t, \pi_{-i}^t)\big) - \big(U_i(\pi^*, a_{-i}^t) - U_i(\pi_i^t, a_{-i}^t)\big)\big| \\
&\le \big|U_i(\pi^*, \pi_{-i}^t) - U_i(\pi^*, a_{-i}^t)\big| - \big|U_i(\pi_i^t, \pi_{-i}^t) - U_i(\pi_i^t, a_{-i}^t)\big| \\
&\le 2.
\end{aligned}$$

Hence, using Azuma-Hoeffding's inequality, for any $\delta \in (0,1)$,

$$\begin{aligned}
1 - \delta &\le \mathbb{P}\left[\sum_{t=1}^T w^t \le \sqrt{8T\log\frac{1}{\delta}}\right] \\
&= \mathbb{P}\left[\left(\sum_{t=1}^T U_i(\pi^*, \pi_{-i}^t) - \sum_{t=1}^T U_i(\pi_i^t, \pi_{-i}^t)\right) - \left(\sum_{t=1}^T U_i(\pi^*, a_{-i}^t) - \sum_{t=1}^T U_i(\pi_i^t, a_{-i}^t)\right) \le \sqrt{8T\log\frac{1}{\delta}}\right] \\
&= \mathbb{P}\left[\sum_{t=1}^T U_i(\pi^*, \pi_{-i}^t) - \sum_{t=1}^T U_i(\pi_i^t, \pi_{-i}^t) \le R_i^T + \sqrt{8T\log\frac{1}{\delta}}\right],
\end{aligned}$$

Since the above expression holds for any $\pi^*$, in particular, using the union bound,

$$\mathbb{P}\left[\max_{\pi^*}\sum_{t=1}^T U_i(\pi^*, \pi_{-i}^t) - \sum_{t=1}^T U_i(\pi_i^t, \pi_{-i}^t) \le R_i^T + \sqrt{8T\log\frac{|\mathcal{A}_i|}{\delta}}\right] \ge 1 - \delta.$$

Summing for $i \in \{1, \dots, N\}$ and using the union bound, we can further write

$$\mathbb{P}\left[\sum_{i=1}^{N} \max_{\pi_i^*} \left\{\sum_{t=1}^{T} U_i(\pi_i^*, \pi_{-i}^t)\right\} - \sum_{t=1}^{T} \sum_{i=1}^{N} U_i(\pi_1^t, \dots, \pi_N^t) \leq \sum_{i=1}^{N} R_i^T + N\sqrt{8T \log \frac{\max_i |\mathcal{A}_i|}{\delta}}\right] \geq 1 - N\delta.$$

Dividing by $T$ and noting that for any player $i \in \{1, \dots, N\}$,

$$\frac{1}{T}\sum_{t=1}^{T} U_i(\pi_i^*, \pi_{-i}^t) = U_i\left(\pi_i^*, \frac{1}{T}\sum_{t=1}^{T} \pi_{-i}^t\right) = U_i\left(\pi_i^*, \bar{\pi}_{-i}^T\right),$$

further yields

$$\mathbb{P}\left[\sum_{i=1}^{N} \max_{\pi_i^*} U_i(\pi_i^*, \bar{\pi}_{-i}^T) - \frac{1}{T}\sum_{t=1}^{T}\sum_{i=1}^{N} U_i(\pi_1^t, \dots, \pi_N^t) \leq \sum_{i=1}^{N} \frac{R_i^T}{T} + N\sqrt{\frac{8}{T} \log \frac{\max_i |\mathcal{A}_i|}{\delta}}\right] \geq 1 - N\delta.$$

We now analyze the term $\Delta := -\frac{1}{T}\left(\sum_{t=1}^{T}\sum_{i=1}^{N} U_i(\pi_1^t, \dots, \pi_N^t)\right)$, which can be expressed as

$$\Delta = -\frac{1}{T}\sum_{t=1}^{T}\sum_{i=1}^{N} U_i(\pi_1^t, \dots, \pi_N^t) = -\frac{1}{T}\sum_{t=1}^{T}\sum_{i=1}^{N} U_i(\pi_1^t, \dots, \pi_N^t) = -\sum_{i=1}^{N} U_i\left(\bar{\pi}_1^T, \dots, \bar{\pi}_N^T\right).$$

Therefore we have

$$\mathbb{P}\left[\sum_{i=1}^{N} \max_{\pi_i^*} \left\{U_i(\pi_i^*, \bar{\pi}_{-i}^T) - U_i(\bar{\pi}_i^T, \bar{\pi}_{-i}^T)\right\} \leq \sum_{i=1}^{N} \frac{R_i^T}{T} + N\sqrt{\frac{8}{T} \log \frac{\max_i |\mathcal{A}_i|}{\delta}}\right] \geq 1 - N\delta.$$

Since $\max_{\pi_i^*} \left\{U_i(\pi_i^*, \bar{\pi}_{-i}^T) - U_i(\bar{\pi}_i^T, \bar{\pi}_{-i}^T)\right\} \geq 0$ for all $i$, the inequality above implies that

$$\mathbb{P}\left[\max_{i \in [N]} \max_{\pi_i^*} \left\{U_i(\pi_i^*, \bar{\pi}_{-i}^T) - U_i(\bar{\pi}_i^T, \bar{\pi}_{-i}^T)\right\} \leq \sum_{i=1}^{N} \frac{R_i^T}{T} + N\sqrt{\frac{8}{T} \log \frac{\max_i |\mathcal{A}_i|}{\delta}}\right] \geq 1 - N\delta.$$

which is equivalent to the statement after making the variable substitution $\delta := \delta'/N$. $\qquad\square$

## A.4 NECESSITY OF ASSUMPTION 1

Recall our definition that a mechanism $M$, under agent strategies $\sigma$, is said to be $(\rho, \gamma)$-approximate if its expected outcome satisfies:

$$\rho\mathbb{E}[U_{M,\sigma}] + \gamma \geq \mathbb{E}[U_{\max}],$$

where $\rho$ and $\gamma$ represent the multiplicative and additive approximation factors respectively. Following Shin et al. (2023), we define the *price of anarchy* (PoA) as a measure of the worst-case efficiency loss due to strategic behavior. Specifically, the multiplicative price of anarchy $\text{PoA}_m$ is the smallest $\rho$ such that the mechanism is $(\rho, 0)$-approximate under every Nash equilibrium. Similarly, the additive price of anarchy $\text{PoA}_a$ is the smallest $\gamma$ such that the mechanism is $(1, \gamma)$-approximate under all equilibria. In contrast, the *price of stability* (PoS) captures the best-case performance at equilibrium: the multiplicative price of stability $\text{PoS}_m$ is the smallest $\rho$ such that there exists some Nash equilibrium under which the mechanism is $(\rho, 0)$-approximate, and the additive price of stability $\text{PoS}_a$ is similarly defined for $(1, \gamma)$-approximation.

**Lemma 9** (Shin et al. (2023)). *Suppose that both the principal's utility functions are supported on $[0, L]$ for some $L > 0$. For any $\varepsilon > 0$, there exists a problem instance such that $\text{PoS}_a \geq L - \varepsilon$, i.e. $\mathbb{E}[U_{M,\sigma}] \leq \varepsilon$.*

**Lemma 10** (Shin et al. (2023)). *With symmetric agents, there exists no PIM such that $\text{PoA}_a < \mathbb{E}\left[U_{\max} - \max_i U_{i,\min}\right]$, where $U_{\max}$ denotes the principal utility of the optimal answer among all candidate answers generated by agents, and $U_{i,\min}$ denotes the principal utility of agent $i$'s worst answer.*

Lemma 9 illustrates a worst-case scenario in which the principal cannot access a good answer. This occurs when a super agent, not aligned with the principal, strategically submits low-utility answers that the principal is forced to accept, resulting in expected utility arbitrarily close to zero—even when much better answers exist.

Lemma 10 indicates that when the utility of agents is negatively correlated with that of the principal, agents tend to act selfishly and adversarially, submitting answers that harm the principal's objective. Introducing independence addresses this issue by decoupling their incentives, thus reducing strategic misalignment. However, in our framework, the principal and agent are reacted based on same questions, therefore, assuming independent utility doesn't make sense. Therefore, these lemmas establish the necessity of the second and third part of Assumption 1.

## B COMPUTATIONAL RESOURCES

Most of the compute is spent in Stage 1 (candidate answers generation). Table 5 shows mean model calls and generated tokens per question to generate candidate answers after 16 roll-outs. Stage 2 (delegation) is comparatively cheap—about 1.511 s per iteration per question. On a single NVIDIA L4, completing 16 rollouts for 300 GSM8K questions takes roughly four days when each agent has access to 250 GB of memory. Runtime scales approximately linearly with the number of questions (and rollouts) and can be reduced substantially by distributing questions across multiple GPUs.

Table 5: Inference cost of generating candidate answers stage of COMMAND on GSM8K: mean model calls and generated tokens per question

|  | Mistral-8B-Instruct | Zephyr-7B | Phi3-Mini-4K-Instruct | Falcon-7B-Instruct |
|---|---|---|---|---|
| Avg. calls | 98.45 | 73.30 | 102.82 | 57.60 |
| Avg. generated tokens | 144.80K | 81.74K | 263.76K | 51.81K |

