# OpenReview forum: "Competitive Multi-Agent Delegation for  LLM Reasoning"
_ICLR.cc/2026/Conference — Submitted to ICLR 2026_

### Official Review · Reviewer_hjLk · 2025-10-31

**Soundness:** 2
**Presentation:** 2
**Contribution:** 2
**Rating:** 4
**Confidence:** 4

**Summary:**

The paper proposes COMMAND, a game-theoretic framework that enhances LLM reasoning through competition among multiple agents evaluated by a principal model. Each agent independently generates candidate answers and receives rewards based on both its internal confidence and the principal’s ranking feedback. Theoretically, the authors prove that multi-agent delegation yields higher expected utility than single-agent setups under fair comparison. Empirically, COMMAND improves accuracy across math benchmarks.

**Strengths:**

- The paper addresses an interesting and relevant problem: how to enhance the reasoning performance of LLMs through competitive multi-agent delegation under a principal–agent framework. The idea of coordinating multiple policies via a principal for self-improvement without explicit fine-tuning is conceptually appealing.
- The framework is supported by theoretical analysis, providing regret bounds and convergence guarantees under online mirror descent, which adds mathematical grounding to the proposed approach.
- The work connects multi-agent learning and game theory with LLM reasoning, an angle that is potentially useful for understanding cooperative-competitive dynamics in large models.

**Weaknesses:**

- Many implementation details are missing or unclear, making it difficult to fully understand or reproduce the method. For example:
  - It is not clearly explained how the principal aggregates responses or computes the global utility mentioned in Line 111.
  - The MCTS process is under-specified: what constitutes a node, how rollouts are defined, and how branching decisions are made remain ambiguous.
  - The paper does not specify the number of iterations in the competitive loop or the stopping criterion for convergence.
- In Table 1, it is unclear which base models are used for the reported baselines. Are they the same as the principal’s model, or different ones? Moreover, since the policies in COMMAND may include stronger models than the principal, the paper should include a baseline reflecting the best single policy’s performance for fair comparison.
- Table 2 and Figure 2 seem to indicate that COMMAND does not outperform the strongest policy in the ensemble and may even slightly underperform it, suggesting that the competitive interaction does not yield consistent gains. I'm not sure whether I misinterpret the results. Correct me if I'm wrong.
- The experimental scope is limited, focusing solely on mathematical reasoning tasks. Broader evaluation on other domains would strengthen the generality of the claims.
- The practical contribution is somewhat limited. Although the theory is sound, the improvement margins are small and the framework lacks clear insights into why competition helps or when it may hurt. Also, the relationship between the consumed computation and the performance of COMMAND and other baselines is not reported. It seems that COMMAND may require significantly more compute than other baselines, while yielding only marginal improvement.
- No ablation or sensitivity analysis is provided to isolate the effects of key components (e.g., number of agents, principal choice). Without this, the contribution feels more conceptual than empirically validated.

**Questions:**

See above

---

> ### Author Response · Authors · 2025-11-29
>
> We extend our sincere appreciation for your valuable feedback and suggestions. Regarding your concerns, we would like to offer further clarification.
>
> **W1 and W2: Missing implementation details**
>
> We thank the reviewer for pointing out that several implementation details were not sufficiently described. We apologize for the missing implementation details and clarify below that all baselines—including Few-shot CoT, rStar, and CoT-SC—use exactly the same backbone LLMs as the agent models in COMMAND. We compare COMMAND against three strong reasoning baselines:1) Single-round CoT prompting, few-shot CoT.  2) Self-Consistency with Chain-of-Thought (CoT-SC) follows widely adopted self-consistency method [1], where we sampled the answers 16 times, employing majority voting for the selection of the answers. 3) rStar is implemented following the template described in the original work [2]. We adopt a simplified variant that only verifies and scores the agents’ submitted answers.
>
> In our experiment. We adopt three open-source instruction-following LLMs to serve as agents in our delegation framework: Mistral-8B-Instruct, Zephyr-7B-Beta and Phi-3-Mini-4K-Instruct. In the candidate answer generation stage, we augment each agent with Monte Carlo Tree Search (MCTS), performing 16 roll-outs with 5 maximum depth. We collect all trajectories and propose mutual reasoning consistency for answer selection. In the delegation stage, We have extended our experiments to include more recent models (e.g. Qwen2/Qwen2.5-7B-Instruct) as principal LLM, providing feedback via best-path masking and consistency checks to the agents. The prompts used throughout the system adhered to the format described in the work by [2].
>
> **W3: On the interpretation of Figure 2 and Table 2**
>
> We thank the reviewer for this thoughtful question. In Table 2 in our submission, our focus is on how individual agents benefit from the competitive interaction under COMMAND: for each agent we report its accuracy after competition, showing that agents’ policies are strengthened through interaction; for COMMAND itself, we report the accuracy of the majority vote over all agents. In contrast, the “strongest policy” corresponds to the best single agent in the ensemble selected post hoc on the test set, which is effectively an oracle baseline that assumes access to labels and therefore is not a realistic test-time strategy. It is thus expected and reasonable that the majority-vote performance of COMMAND may be slightly below this oracle “strongest policy,” because in practice we cannot know in advance which agent will perform best and must aggregate agents’ outputs in a label-free manner, for which majority vote is a standard and robust choice. Figure 2 then tracks how COMMAND improves the accuracy over iterations. We will clarify this evaluation protocol in the revised version, explicitly noting that (i) Table 2 reports per-agent performance after competition, (ii) Figure 2 focuses on the evolution of majority-vote accuracy under COMMAND, and (iii) the “strongest policy” should be interpreted as an oracle upper bound rather than a deployable baseline.
>
> **W4: Scope of Experimental Evaluation**
>
> We thank the reviewer for this helpful suggestion and fully agree that broader evaluation beyond mathematical reasoning would further strengthen the empirical part of the paper. In this work, we deliberately focused on math benchmarks as a first testbed because they provide (i) well-established, automatically verifiable ground truth and (ii) a clean, controlled setting to study multi-step reasoning and our game-theoretic mechanism. Given time and resource constraints, we were not able to extend the empirical study to additional domains, but we see this as a natural next step and will add a discussion in the revision outlining how COMMAND can be instantiated for code generation, planning, or open-ended QA. Importantly, our theoretical results are task-agnostic: the guarantees depend only on the properties of the scoring rule and the interaction dynamics between principal and agents, not on any assumption specific to mathematical problems, so extending the experiments to other domains does not affect the validity of the theory.
>
> [1] Wang, Xuezhi, et al. "Self-Consistency Improves Chain of Thought Reasoning in Language Models." ICLR.
> [2] Qi, Z., Mingyuan, M. A., Xu, J., Zhang, L. L., Yang, F., & Yang, M. Mutual Reasoning Makes Smaller LLMs Stronger Problem-Solver. ICLR.

---

> ### Author Response · Authors · 2025-11-29
>
> **W5: Practical Impact and Computational Resources**
>
> We thank the reviewer for these thoughtful comments. While the absolute gains over strong baselines may appear modest, they are obtained on challenging math benchmarks where existing baselines are already highly optimized, so consistent 1–3\% improvements are practically meaningful, especially given that COMMAND is training-free without fine-tuning or RLHF.
> As detailed in Appendix B, most of the computational cost occurs in Stage 1, where candidate answers are generated. Table 1 reports the mean number of model calls and generated tokens per question required to obtain candidate trajectories under 16 roll-outs. This cost scales linearly with both the number of questions and the number of roll-outs. In contrast, Stage 2 (delegation) is comparatively inexpensive - approximately 1.511 seconds per iteration per question. No additional infrastructure is needed to coordinate multiple agents.
>
> Importantly, unlike traditional fine-tuning pipelines that require hours to days of compute, the COMMAND mechanism is fully training-free. Full-model fine-tuning typically demands thousands of GPU hours and can cost between $10,000$ and over $35,000$ per run [3]. Even parameter-efficient fine-tuning (PEFT) approaches such as FinLoRA [4] still require several hours to days to update model weights. In contrast, COMMAND operates without modifying model parameters or performing any gradient updates, thereby eliminating the need for a training phase entirely.
>
> **Table 1: Inference cost of generating candidate answers in the COMMAND stage on GSM8K
> (mean model calls and generated tokens per question)**
>
> |                          | **Mistral-8B-Instruct** | **Zephyr-7B** | **Phi3-Mini-4K-Instruct** | **Falcon-7B-Instruct** |
> |:------------------------:|:------------------------:|:--------------:|:--------------------------:|:------------------------:|
> | **Avg. calls**           | 98.45                   | 73.30         | 102.82                    | 57.60                   |
> | **Avg. generated tokens**| 144.80K                 | 81.74K        | 263.76K                   | 51.81K                  |
>
> **W6: Sensitivity analysis**
>
> We thank the reviewer for highlighting the importance of ablations and sensitivity analysis. We conducted an additional set of experiments in which we replaced the original principal with Qwen2.5-7B-Instruct in Table 2, while keeping the pool of execution agents unchanged. The new results show that the relative gains of COMMAND over the baselines remain stable across all three math benchmarks. This suggests that our mechanism is robust to the specific choice of principal, as long as the principal provides a reasonably informative reward signal.
>
> **Table 2: Accuracy (%) across benchmark datasets for each method.**
> |**Dataset**|**Few-shot CoT**|**CoT-SC@maj16**|**Principal**|**rStar**|**COMMAND**|
> |:-----------:|:----------------:|:----------------:|:-------------:|:---------:|:-----------:|
> |Math|8.0|13.2|10.2|33.0| **34.8**|
> |GSM8K|40.4|62.8|33.5|57.1| **64.3** |
> |GSM-Hard|17.7|24.1|15.6| 22.8| **24.6**|
>
> Once again, we sincerely appreciate the reviewer’s time and thoughtful feedback, and we hope our responses help clarify any remaining concerns.
>
> [3] Liu A, Feng B, Xue B, et al. Deepseek-v3 technical report[J]. arXiv preprint arXiv:2412.19437, 2024.
> [4] Wang, D., Patel, J., Zha, D., Yang, S. Y., & Liu, X. Y. (2025). FinLoRA: Benchmarking LoRA Methods for Fine-Tuning LLMs on Financial Datasets. arXiv preprint arXiv:2505.19819.

---

### Official Review · Reviewer_CTzi · 2025-10-31

**Soundness:** 3
**Presentation:** 3
**Contribution:** 1
**Rating:** 2
**Confidence:** 5

**Summary:**

The paper proposes COMMAND, a training-free framework that uses game-theoretic principles to improve LLM reasoning through competitive multi-agent delegation. In this setup, a principal LLM ranks answers submitted by multiple agent LLMs. Each agent's utility function combines two components: its internal confidence (measured by self-consistency across its own samples) and the principal's ranking feedback. Agent policies are then updated using mirror descent with the Hedge algorithm.
The authors make three main theoretical claims: (i) multi-agent delegation provably outperforms single-agent approaches under fair candidate budgeting, (ii) each agent achieves sublinear regret, and (iii) the time-averaged policies converge to an approximate Nash equilibrium. Empirically, COMMAND shows accuracy improvements on MATH, GSM8K, and GSM-Hard compared to few-shot CoT, a simplified r* (rStar) variant, and a "Principal-alone" baseline (Sections 1–3, Section 4.2 Table 1, Figure 2).
However, I notice this work appears quite similar to https://arxiv.org/abs/2506.08292—I'd appreciate the authors clarifying the relationship.

**Strengths:**

Motivated game formulation. The "delegation game" design is elegant: agents optimize a utility that combines their own self-consistency signal with the principal's ranking feedback. This explicitly aligns each agent's search process with the principal's evaluation criteria. The ranking-based reward structure (top=+1, bottom=−1) and the mirror-descent policy updates using exponential weights provide a concrete, implementable mechanism that requires no fine-tuning (Sections 2.2–2.3, Algorithm 1).

Theoretical foundations built. The analysis builds on well-established assumptions (Pareto-optimal play, agent symmetry, non-negative alignment) and standard online learning theory. Theorem 1 formalizes why delegation outperforms single-agent approaches under equal candidate budgets. Theorem 2 proves O(√T) regret with learning rate η=1/√T. Theorem 3 establishes that time-averaged policies converge to a ξT(δ)-approximate Nash equilibrium. Together, these results provide COMMAND with a principled theoretical backbone (Sections 3.1–3.3).

Reasonable experimental setup. The evaluation uses heterogeneous 7–8B parameter agents (Mistral-8B-Instruct, Zephyr-7B-Beta, Phi-3-Mini-Instruct, Falcon-7B-Instruct) with LLaMA-2-7B as the principal. Candidate generation employs MCTS with 16 rollouts at depth 5. The benchmarks span MATH, GSM8K, and GSM-Hard with clearly reported sample counts of 300/300/320 respectively (Section 4.1).

**Weaknesses:**

Significant overlap with concurrent work—this is my primary concern. The closest contemporary work is ECON (From Debate to Equilibrium, arXiv:2506.08292), which is not cited in the paper. ECON also formulates multi-LLM coordination as a game and seeks a (Bayesian) Nash equilibrium with regret guarantees. While ECON uses a hierarchical RL procedure rather than training-free mirror descent, it reports 11.2% mean gains across six reasoning and planning benchmarks (ICML 2025). Given this substantial overlap in problem formulation and approach, I have concerns about the novelty of the contribution, which has influenced my score.

Theory-practice gap in symmetry assumptions. Assumption 1-ii requires symmetric agents sampling from "the same distribution D" (Section 3.1, page 4). However, the experimental agents come from different model families (Mistral, Zephyr, Phi-3, Falcon) with inherently different sampling distributions. While the paper argues these models have "comparable capacity" and use "identical sampling procedures," this doesn't satisfy the formal symmetry requirement. This gap weakens the applicability of Theorem 1's theoretical comparison to the experimental results (Section 4.1, page 6).

Missing critical baseline for Theorem 1. Theorem 1's central claim compares single-agent versus multi-agent performance under equal total candidate budgets. However, the empirical "Principal" baseline doesn't appear to use the same total number of candidates as the multi-agent system (where each agent runs 16 MCTS rollouts). The paper doesn't report a "single-agent with the same total candidate pool" ablation, so the core theoretical prediction isn't directly validated experimentally (Sections 3.1, 4.1–4.2).

Limited evaluation scope. All tasks are math-centric; there's no evaluation on code generation, planning, or open-ended QA where verification is more challenging. The evaluation uses relatively small subsets (300/300/320 examples) without reporting confidence intervals or significance tests. Additionally, rStar is implemented as a simplified verifier-only variant, which may not represent a strong baseline (Sections 4.1–4.2).

**Questions:**

Direct test of Theorem 1. Could you add a single-agent baseline that receives the same total candidate budget as the multi-agent system? For example, one agent could select from the union of all candidates produced by the multi-agent pool. This would directly test Theorem 1's theoretical setup (Sections 3.1, 4.2).

Connection to Bayesian equilibrium. Is there an interpretation of your mirror-descent updates as seeking a (Bayesian) equilibrium under uncertainty? This might help clarify the relationship to ECON.

Robustness to violated symmetry. What happens when Assumption 1-ii (symmetry) is violated, as it is with your heterogeneous agents? Do you have any theoretical extensions or empirical ablations studying scenarios where agent utilities come from different distributions? (Sections 3.1, 4.1)

Principal model sensitivity. How sensitive are the results to the choice of principal? If you swap to a different model family (e.g., Mistral or Llama-3) or use a verifier-based reward, do both the absolute accuracy and relative gains change significantly? (Section 4.1)
Generalization beyond math. Can COMMAND handle tasks without easily verifiable solutions, such as planning, code synthesis, or open-ended QA? Do you have any preliminary results beyond mathematical reasoning? (Sections 4.1–4.2)

---

> ### Author Response · Authors · 2025-11-29
>
> We extend our sincere appreciation for your valuable feedback and suggestions. Regarding your concerns, we would like to offer further clarification.
>
> **W1 and Q2: Significant overlap with concurrent work (ECON)**
>
> We sincerely thank the reviewer for pointing out this important related work and apologize for our oversight in not citing it. We will add ECON and discuss it explicitly in the revised version. We agree that, at a high level, both approaches share the spirit of using feedback and iterative refinement for multi-agent reasoning, but there are substantial differences in motivation, mechanism, and theoretical guarantees.
>
> First, the motivation behind ECON is multi-agent coordination, modeled through mutual beliefs among agents. Each execution LLM is equipped with a belief network that maps its local trajectory and current observation to a latent belief state and a prompt-control strategy, and the reward explicitly embeds a consensus component (e.g., an action-likelihood term that rewards agreement with the collective solution). In contrast, our framework leverages ranking-based feedback from a principal LLM to create a competetive environment: agents do not communicate with each other, and are incentivized to provide diverse reasoning and better responses purely through relative ranking, rather than explicit coordination or shared beliefs.
>
> Second, at the implementation level, ECON belongs to the multi-agent RL paradigm: it relies on a value function and temporal-difference updates, with additional networks that must be trained and tuned. Our method is entirely training-free: we apply mirror-descent style updates directly in the space of agent strategies using only ranking feedback, without learning auxiliary networks or fine-tuning the underlying LLMs. This makes our mechanism lightweight and easy to plug into existing systems.
>
> Third, on the theoretical side, ECON analyzes Bayesian Nash equilibria and Bayesian regret under its belief-based RL procedure, whereas our analysis focuses on prior-independent, feedback-driven learning dynamics. We prove an $O(\sqrt{T})$ regret bound for our mirror-descent updates and show convergence to an equilibrium policy profile under our ranking game, ensuring that the process not only approaches but actually converges to a stable policy (rather than potentially wandering around an equilibrium set). This convergence guarantee under a prior-free, ranking-based mechanism is, to our knowledge, not covered by ECON.
>
> Finally, regarding empirical results, we agree that ECON reports an average $11.2\\%$ improvement across six benchmarks, but this aggregate number mixes heterogeneous datasets and model sizes, making direct comparison difficult. Focusing on a controlled setting with the Mistral-7B backbone on math-reasoning benchmarks (GSM8K, GSM-Hard, MATH), ECON achieves an average $1.9\\%$ gain over rStar (Figures 9-11 in ECON), while COMMAND attains a $3.6\\%$ improvement over rStar. Under this comparable setup, our framework shows competitive and even stronger gains.
>
> We also greatly appreciate the reviewer’s question about the connection to Bayesian equilibria under uncertainty. In standard Bayesian game-theoretic formulations, each agent $i$ is endowed with a belief model over an unknown state $\theta$ (or, equivalently, over the principal’s scoring rule) and chooses an action to maximize its Bayesian expected payoff
> $\mathbb{E}_{\theta}[u\_i(a\_i, a\_{-i}; \theta)]$.
> By contrast, our setting emphasizes a competitive environment that incentivizes agents’ reasoning, and we do not assume any explicit prior or belief model; instead, we work with a prior-independent mechanism driven purely by observed feedback. Our analysis relies only on payoff signals and the no-regret property of mirror descent, making the mechanism robust to misspecified or unknown beliefs.
> From this perspective, the mirror-descent updates can be viewed as a feedback-driven learning process that approximates a Bayesian-style equilibrium: agents iteratively adjust their policies based on feedback that depends on both their own actions and those of others, and our no-regret guarantees ensure that, over time, the joint play converges to an equilibrium policy profile. Thus, although we do not impose an explicit Bayesian belief structure, our framework captures a similar notion of adaptation to uncertainty through learning, while remaining prior-independent and competitive in spirit.

---

> > ### Author Response · Authors · 2025-11-29
> >
> > **W2: Theory-practice gap in symmetry assumptions**
> >
> > We thank the reviewer for this careful reading of Assumption 1(ii). Our theoretical analysis assumes that agents are symmetric in the sense that their actions are drawn from the same distribution $D$. We agree that our empirical agents (Mistral, Zephyr, Phi-3, Falcon) are not literally identical instantiations of the same model, so this assumption is stricter than our main experimental setup. To further narrow this theory–practice gap, we conducted a new experiment in which all execution agents are identical: all agents are Mistral-7B-Instruct (with different sampling temperatures), and the principal is Qwen/Qwen2.5-7B-Instruct. The results in Table 1 report the math accuracy for all baselines, while Table 2 reports, for each agent, its accuracy before and after applying COMMAND. We find that COMMAND still consistently improves accuracy when every agent is identical, and that each individual agent benefits from COMMAND, confirming that COMMAND enhances both collective reasoning and the reasoning quality of a single underlying model. Together, this new identical-agent experiment strengthen the applicability of Theorem 1 to our empirical results.
> >
> > **Table 1: Accuracy (%) across MATH dataset for the same agent.**
> > |**Dataset**|**Few-shot CoT**|**CoT-SC@maj16**|**rStar**|**COMMAND**|
> > |:-----------:|:----------------:|:----------------:|:---------:|:-----------:|
> > |Math|8.6|13.2|33.0|**34.4**|
> >
> > **Table 2: Accuracy (%) of Mistral before and after applying COMMAND for MATH dataset.**
> > |**Agent**|**Before**|**After**|
> > |:------------------------:|:----------:|:---------:|
> > |Mistra_01 (temp = 0.6)|33.4| 34.2|
> > |Mistra_02 (temp = 0.8)|30.4| 31.4|
> > |Mistra_03 (temp = 0.9)|33.0| 33.6|
> >
> >
> > **W3 and Q1: Single-Agent Equal-Budget Baseline for Theorem 1**
> >
> > We thank the reviewer for highlighting this important point. Theorem 1 analyzes a comparison in which a single agent and a multi-agent system are given the same total candidate budget. We agree that our original experimental baseline did not explicitly enforce this total-budget matching. Due to the limited time and computational budget available during the rebuttal period, we were unfortunately not able to re-run experiments with this new baseline, as we need to use MCTS to generate new candidate answers. Nevertheless, we fully agree that this is an important and fair comparison, and we will include this principal–single-agent pooled-candidate baseline, together with its results in the revised version.
> >
> > **W4 and Q4: Evaluation scope and principal sensitivity**
> >
> > We thank the reviewer for these valuable comments on our experimental design. We agree that using only partial test sets may limit the completeness of the evaluation. In the revised version, we now evaluate COMMAND on the full GSM8K, GSM-Hard, and MATH500 test sets. The updated results are shown in Table 3. Across all three benchmarks, COMMAND consistently achieves the best performance, improving over rStar by 1.8–3.6 points in absolute accuracy and substantially outperforming Few-shot CoT and CoT-SC. These gains closely match those observed on the original subsets, indicating that our conclusions remain valid when evaluated on the full test sets.
> >
> > **Table 3: Accuracy (%) across benchmark datasets for each method.**
> > |**Dataset**|**Few-shot CoT**|**CoT-SC@maj16**|**Principal**|**rStar**|**COMMAND**|
> > |:-----------:|:----------------:|:----------------:|:-------------:|:---------:|:-----------:|
> > |Math|8.0|13.2|10.2|33.0| **34.8**|
> > |GSM8K|40.4|62.8|33.5|57.1| **64.3** |
> > |GSM-Hard|17.7|24.1|15.6| 22.8| **24.6**
> >
> > We also agree that extending beyond math-centric tasks to code generation, planning, and open-ended QA is important, especially where verification is more challenging. In this work, we deliberately focus on math reasoning because it offers (i) well-established, automatically verifiable benchmarks and (ii) a clean setting to test multi-step reasoning and our game-theoretic mechanism. Our theoretical framework, however, is task-agnostic: Theorem 1 and the analysis in Section 3 only assume access to a utility function over candidate answers and do not rely on any math-specific structure. Due to time and computational constraints for this submission, we were not able to include additional task families, and we will explicitly state this limitation and highlight code, planning, and open-ended QA as promising directions for future empirical validation.
> >
> > Finally, to assess the sensitivity of COMMAND to the choice of principal model, we conducted an additional set of experiments in which we replaced the original principal with Qwen2.5-7B-Instruct in Table 3 while keeping the pool of execution agents unchanged. The new results show that the relative gains of COMMAND over the baselines remain stable across all three math benchmarks. This suggests that our mechanism is robust to the specific choice of principal, as long as the principal provides a reasonably reward signal.

---

> > > ### Author Response · Authors · 2025-11-29
> > >
> > > **Q3: Robustness when symmetry is violated**
> > >
> > > We thank the reviewer for asking about robustness beyond Assumption 1(ii). Our result in Theorem 1 relies on agents being symmetric, and in the fully heterogeneous case we show in Appendix A.4 that some form of symmetry is indeed necessary. More concretely, when both the principal’s and the agents’ utilities are bounded, we can construct a problem instance in which a misaligned “super agent’’ causes the principal to select an extremely poor answer, even though much better answers exist. This worst-case example corresponds to a heterogeneous setting where a super agent, whose preferences are not aligned with the principal, strategically submits low-utility answers that the principal is effectively forced to accept. Lemma 9 thus illustrates that once Assumption 1(ii) is dropped entirely, the principal’s performance can become arbitrarily bad; some approximate symmetry is required for any meaningful upper bound. Empirically, when there is a single, significantly more accurate agent, it tends to be selected almost all the time, leaving little room for other agents to update and preventing a genuinely competitive environment from emerging.
> > >
> > > Once again, we sincerely appreciate the reviewer’s time and thoughtful feedback, and we hope our responses help clarify any remaining concerns.

---

### Official Review · Reviewer_A5qW · 2025-11-01

**Soundness:** 2
**Presentation:** 2
**Contribution:** 2
**Rating:** 4
**Confidence:** 4

**Summary:**

This paper introduces COMMAND, a game-theoretic framework for improving LLM reasoning through competitive multi-agent delegation. In this framework, a principal LLM assigns reasoning tasks to multiple agent LLMs that generate candidate answers and compete for rewards. Each agent's utility combines its internal confidence with the principal's ranking-based evaluation, incentivizing both high-quality outputs and alignment with the principal's preferences. Empirical evaluations on GSM8K, MATH, and GSM-Hard demonstrate modest accuracy improvements over baselines.

**Strengths:**

1. The framework is training-free, requiring no fine-tuning or parameter updates, and uses only inference-time computation.

2. The paper provides three theorems with complete proofs establishing that multi-agent systems can outperform single-agent counterparts.

3. Experiments show  gains in mathematical reasoning compared to single-agent baselines.

**Weaknesses:**

1. The paper claims "under fair comparison, multi-agent systems outperform their single-agent counterparts". However, recent work "Debate or Vote: Which Yields Better Decisions in Multi-Agent Large Language Models?" has shown that Majority Voting accounts for most performance gains in multi-agent systems, and proved theoretically that debate alone does not improve expected correctness. Therefore, I have two concerns:
- How do the authors clarify the contradiction between their theory and recent theoretical results?
- Is the competitive delegation mechanism adding value beyond simple aggregation? Without comparisons to majority voting baselines, it is unclear whether the gains stem from the game-theoretic mechanism or simply from having more independent samples to aggregate.

2. The experiments use relatively old and weak models, such as LLaMA-2-7B. These models have limited reasoning capabilities, making the experiments less convincing. The paper should validate the approach on stronger, more recent models such as Qwen3.

3. All experiments focus exclusively on mathematical reasoning tasks. To substantiate claims about general multi-agent LLM reasoning, the paper should evaluate on diverse domains and standard benchmarks such as MMLU, HumanEval, and HellaSwag. The current narrow evaluation severely limits the generalizability of the findings.

4. The paper uses Monte Carlo Tree Search, but its focus is on the advantages of a multi-agent system over a single agent. Therefore, comparing MCTS + multi-agent with a single agent is unfair and requires ablation experiments. However, I did not see any relevant ablation experiments in the paper, making it difficult to believe that the performance improvement comes from multi-agent.

5. The paper's presentation makes it easy for readers to get lost. And the paper missed some critical details, such as baseline implementations (Which LLM does each baseline use?). These details are essential for reproducibility and fair comparison.

**Questions:**

The paper does not clearly specify which LLM is used for baselines e.g., Few-shot CoT. Are they using LLaMA-2-7B-Instruct? This information is critical for fair comparison.

---

> ### Author Response · Authors · 2025-11-29
>
> We extend our sincere appreciation for your valuable feedback and suggestions. Regarding your concerns, we would like to offer further clarification.
>
> **W1: Clarification on "Majority Voting accounts for most performance gains in multi-agent systems"**
>
> We thank the reviewer for pointing out this important related work and the potential contradiction. The paper “Debate or Vote: Which Yields Better Decisions in Multi-Agent Large Language Models?” analyzes a different setting from ours. Their theoretical and empirical analysis compares aggregation rules within multi-agent systems (e.g., debate vs. majority vote), and shows that, under their assumptions, majority voting captures most of the performance gains. Our work instead focuses on a different question: whether multi-agent systems outperform single-agent counterparts under a fixed number of answers and a fixed computational budget. In other words, we compare multi-agent vs. single-agent systems, while the previous paper compares different decision rules within a multi-agent setup. Thus, although both works study multi-agent LLMs, the comparison settings and objectives are not directly comparable.
>
> Moreover, the optimality of majority voting in the previous work is derived in the limit of infinitely many answers and under the assumption that the correct answer has the largest prior belief [1]. This assumption is reasonable only on relatively simple benchmarks. Indeed, in Table 1 from [1], most datasets are relatively easy, with single-model accuracy exceeding 80\%. However, on more challenging benchmarks such as MMLU and HH-RLHF, where accuracy drops around 50\%, the same paper reports that multi-agent debate can outperform majority voting. In these harder regimes, the correct answer may no longer be the most frequently produced response, while majority voting inherently favors the most common (most probable) answer. In such cases, majority voting can fail to recover the correct solution, which is precisely the regime we are interested in.
>
> In our setting, although naive agent debate alone may not always improve the answer, our framework is not based on debate in isolation. Instead, COMMAND incorporates feedback from a principal LLM and introduces a competitive environment through a ranking-based mechanism. Even if the correct answer is not the most favorable among agents, the ranking feedback allows the principal to progressively steer the system toward its preferred (and often more accurate) solution. This idea is formalized in our game design by allowing misalignment between the principal LLM and the agent LLMs, so that an agent’s preferred answer may differ from the principal’s preference. Our Theorem 1 shows that, despite these heterogeneous preferences, the mechanism can still achieve a desirable outcome at equilibrium. Theorems 2 and 3 further establish that our learning dynamics converge to this equilibrium.
>
> We agree that it is important to compare against majority-voting style baselines to isolate the effect of our mechanism from simple aggregation. Therefore, we extend our baselines to include Self-Consistency with Chain-of-Thought (CoT-SC) [2] where we sample 16 candidate answers and select the final prediction by majority vote. These stronger baselines are now included in Table 1. Across all three benchmarks, COMMAND consistently achieves the best performance, improving over rStar by 1.8–3.6 absolute accuracy points and substantially outperforming Few-shot CoT and CoT-SC. Taken together with the evidence from [1] on challenging benchmarks, these results suggest that mechanisms beyond pure majority voting—such as debate and our competitive delegation scheme—are particularly beneficial in the hard regimes where the correct answer is not the most favored response.
>
> **Table 1: Accuracy (%) across benchmark datasets for each method.**
> |**Dataset**|**Few-shot CoT**|**CoT-SC@maj16**|**Principal**|**rStar**|**COMMAND**|
> |:-----------:|:----------------:|:----------------:|:-------------:|:---------:|:-----------:|
> |Math|8.0|13.2|10.2|33.0| **34.8**|
> |GSM8K|40.4|62.8|33.5|57.1| **64.3** |
> |GSM-Hard|17.7|24.1|15.6| 22.8| **24.6**|
>
> [1] Choi, H. K., Zhu, J., & Li, S. Debate or Vote: Which Yields Better Decisions in Multi-Agent Large Language Models?. NeurIPS.
> [2] Wang, Xuezhi, et al. "Self-Consistency Improves Chain of Thought Reasoning in Language Models." ICLR.

---

> > ### Author Response · Authors · 2025-11-29
> >
> > **W2: Experiments use older/weaker models; should include Qwen3 or stronger models**
> >
> > We thank the reviewer for this valuable comment. To address this concern, we have extended our experiments to include more recent models (e.g. Qwen2/Qwen2.5-7B-Instruct) as principal LLM. Our results ub Table 1 show that the choice of principal model does not significantly affect the relative gains provided by COMMAND: upgrading the principal improves absolute accuracy, but the improvement of COMMAND over its corresponding principal baseline remains stable. Due to limited time and computational resources, we did not upgrade all agent models to the latest backbones (e.g., Qwen3). Nevertheless, the observed consistency of COMMAND’s improvements across different principal strengths suggests that our conclusions are not an artifact of using older/weaker models.
> >
> > **W3: Scope of evaluation and generality of findings**
> >
> > We thank the reviewer for highlighting this limitation. Our current experiments indeed focus on mathematical reasoning, which we chose because they are standard, challenging benchmarks that focus on multi-step, open-ended reasoning, the problems where we expect multi-agent interaction and competitive delegation to be most informative. We agree that this narrow focus limits the immediate empirical evidence for fully general multi-agent LLM reasoning. However, our theoretical guarantees are task-agnostic: the analysis is formulated for generic decision problems and applies to any domain where one can define a reward for candidate answers and provide ranking-based feedback from the principal. Benchmarks such as MMLU, HumanEval, and HellaSwag differ not only in domain but also in format (e.g., multiple-choice for MMLU, code-generation for HumanEval), and adapting COMMAND to these settings requires non-trivial changes to prompting, rewarding, and evaluation pipelines that are beyond the time and computational budget of this revision. Thus, while our empirical study is instantiated on mathematical reasoning, the underlying mechanism and theoretical guarantees extend in principle to a broad range of multi-agent LLM reasoning tasks beyond mathematics.
> >
> > **W4: Role of MCTS and fairness of the multi-agent comparison**
> >
> > We thank the reviewer for raising this important concern. We note that our strongest baselines already make use of the same MCTS-style search as COMMAND: Few-shot CoT, rStar-CoT, and CoT-SC@maj16 are all implemented with MCTS, so our main comparisons are not between “multi-agent + MCTS’’ and a plain single agent, but between different reasoning schemes under a shared search procedure and similar compute budgets.
> >
> > **W5 and Q1: Clarity of presentation and missing baseline details**
> >
> > We thank the reviewer for these helpful comments. We apologize for the missing implementation details in the original submission and provide the clarifications below. Importantly, all baselines—including Few-shot CoT, rStar, and CoT-SC—use exactly the same backbone LLMs as the agent models in COMMAND. We compare COMMAND against three strong reasoning baselines:1) Single-round CoT prompting, few-shot CoT [2].  2) Self-Consistency with Chain-of-Thought (CoT-SC) follows widely adopted self-consistency method, where we sampled the answers 16 times, employing majority voting for the selection of the answers. 3) rStar is implemented following the template described in the original work. We adopt a simplified variant that only verifies and scores the agents’ submitted answers.
> >
> > In our experiment. We empoly three open-source instruction-following LLMs to serve as agents in our delegation framework: Mistral-8B-Instruct, Zephyr-7B-Beta and Phi-3-Mini-4K-Instruct. In the candidate answer generation stage, we augment each agent with Monte Carlo Tree Search (MCTS), performing 16 roll-outs with 5 maximum depth. We collect all trajectories and propose mutual reasoning consistency for answer selection. In the delegation stage, We have extended our experiments to include more recent models (e.g. Qwen2/Qwen2.5-7B-Instruct) as principal LLM, providing feedback via best-path masking and consistency checks to the agents. The prompts used throughout the system adhered to the format described in the work by [3].
> >
> > Once again, we sincerely appreciate the reviewer’s time and thoughtful feedback, and we hope our responses help clarify any remaining concerns.
> >
> > [3] Qi, Z., Mingyuan, M. A., Xu, J., Zhang, L. L., Yang, F., & Yang, M. Mutual Reasoning Makes Smaller LLMs Stronger Problem-Solver. ICLR.

---

### Official Review · Reviewer_Rjhr · 2025-11-01

**Soundness:** 2
**Presentation:** 2
**Contribution:** 3
**Rating:** 4
**Confidence:** 4

**Summary:**

This paper proposes COMMAND, a training-free method for multi-agent LLM reasoning based on competitive delegation. Multiple agent LLMs generate candidate answers and compete for rewards determined by a principal LLM's ranking, combined with their internal confidence. The method employs game theory with agents updating policies via online mirror descent to reach the Nash equilibrium. Experiments on GSM8K, MATH, and GSM-Hard show performance improvements over baselines, though gains are modest (2-9%). The paper provides theoretical guarantees for convergence and regret bounds.

**Strengths:**

This paper provides theoretical guarantees showing that the multi-agent framework of COMMAND improves over its single-agent counterpart.

Unlike RL-based approaches or fine-tuning methods, COMMAND works purely at inference time, making it practical for immediate deployment without requiring additional training resources.

Tables 3-4 validate key aspects of Assumption 1, with ~90% Pareto-optimal play compliance and positive correlation between principal and agent utilities (0.15-0.50).

**Weaknesses:**

The paper's evaluation setting is quite odd. The evaluation only used 300 questions from the GSM8K and MATH datasets and 320 questions from the GSM-Hard dataset. In fact, the complete MATH500 test set only has 500 questions, which wouldn't introduce significant computational overhead. Furthermore, the reasoning chains in GSM8K are not very long, and the reviewer considered the computational overhead to be completely acceptable. Conducting experiments on the complete test set will be more convincing.

The paper compares against only three baselines (Few-shot CoT, rStar, Principal), missing several essential comparisons: multi-agent debate, self-consistency, etc.

The paper should include an ablation where all agents use the same LLM (e.g., all agents are LLaMA-2-7B-Instruct). Without this, it is impossible to determine whether performance gains come from the multi-agent mechanism or simply from having one stronger model (e.g., Mistral) in the agent pool.

**Questions:**

NA

---

> ### Author Response · Authors · 2025-11-29
>
> Thank you for your insightful feedback on our manuscript. We appreciate your comments and have taken them into consideration to improve our paper. Below, we do our best to address all concerns adequately.
>
> **W1 and W2: Clarification on evaluation setting and Missing Baselines**
>
> We thank the reviewer for raising this concern. We now evaluate COMMAND on the complete GSM8K, GSM-Hard, and MATH500 test set. The updated results are shown in Table 1. We also extended the baselines to include Self-Consistency with Chain-of-Thought [1], where we sampled the answers 16 times, employing majority voting for the selection of the answers. These stronger baselines are now included in the revised Table below. Across all three benchmarks, COMMAND consistently achieves the best performance, improving over rStar by 1.8–3.6 absolute accuracy  and substantially outperforming Few-shot CoT and CoT-SC. These gains closely match those observed on the original subsets, indicating that our conclusions remain valid when evaluated on the full test sets.
>
> **Table 1: Accuracy (%) across benchmark datasets for each method.**
> |**Dataset**|**Few-shot CoT**|**CoT-SC@maj16**|**Principal**|**rStar**|**COMMAND**|
> |:-----------:|:----------------:|:----------------:|:-------------:|:---------:|:-----------:|
> |Math|8.0|13.2|10.2|33.0| **34.8**|
> |GSM8K|40.4|62.8|33.5|57.1| **64.3** |
> |GSM-Hard|17.7|24.1|15.6| 22.8| **24.6**|
>
> **W3: Ablation with homogeneous agents**
>
> We thank the reviewer for raising this important point. To directly test whether COMMAND’s gains come from multi-agent systems, we conducted a new experiment where all agents  are Mistral-7B-Instruct by setting three different temperatures (0.6, 0.8, and 0.9), and Qwen/Qwen2.5-7B-Instruct as the principal. The results in Table 2 report the  accuracy for all baselines, while Table 3 reports the accuracy before and after applying COMMAND for each agent. COMMAND still consistently improves accuracy even when every agent is identical, demonstrating that heterogeneity is not required for performance gains. Each individual agent benefits from COMMAND, confirming that COMMAND enhances both collective reasoning and single-model reasoning quality. Overall, these results suggest that COMMAND’s gains are driven by the coordination mechanism itself, not merely by including a stronger model in the agent pool.
>
> **Table 2: Accuracy (%) across MATH dataset for the same agent.**
>
> |**Dataset**|**Few-shot CoT**|**CoT-SC@maj16**|**rStar**|**COMMAND**|
> |:-----------:|:----------------:|:----------------:|:---------:|:-----------:|
> |Math|8.6|13.2|33.0|**34.4**|
>
> **Table 3: Accuracy (%) of Mistral before and after applying COMMAND for MATH dataset.**
>
> |**Agent**|**Before**|**After**|
> |:------------------------:|:----------:|:---------:|
> |Mistra_01 (temp = 0.6)|33.4| 34.2|
> |Mistra_02 (temp = 0.8)|30.4| 31.4|
> |Mistra_03 (temp = 0.9)|33.0| 33.6|
>
> Once again, we sincerely appreciate the reviewer’s time and thoughtful feedback, and we hope our responses help clarify any remaining concerns.
>
> [1] Wang, Xuezhi, et al. "Self-Consistency Improves Chain of Thought Reasoning in Language Models." ICLR.

---

### Meta-Review · Area_Chair_SbDF · 2026-01-03

**Summary:**

This paper proposes a training-free method, COMMAND, for multi-agent LLM reasoning based on competitive delegation. All reviewers agree that the paper lacks essential baselines and ablation studies, such as the number of agents, principal choice. Also, the contribution is weakened by overlap with concurrent ECON-style work despite the rebuttal and added experiments.

**Reviewer Concerns:**

Interesting but not sufficiently novel or empirically convincing at this stage, lacks experiments including essential baselines and ablation studies, such as the number of agents and principal choice.

**Reviewer Scores:**

Rjhr: Likely unchanged (4).

A5qW: Likely unchanged (4).

CTzi: Likely unchanged (2).

hjLk: Likely unchanged (4).

---

### Decision · Program_Chairs · 2026-01-26

Reject